



# FTIR time series of tropospheric HCN in eastern China: seasonality, interannual variability and source attribution

Youwen Sun [1], Cheng Liu [1, 2, 3, 4, 5]*, Lin Zhang [6]*, Mathias Palm [7], Justus Notholt [7], Hao Yin [1], Corinne Vigouroux [8], Erik Lutsch [9], Wei Wang [1], Changong Shan [1], Thomas Blumenstock [10], Tomoo Nagahama [11], Isamu Morino [12], Emmanuel Mahieu [13], Kimberly Strong [9], Bavo Langerock [8], Martine De Maziere [8], Qihou Hu [1], Huifang Zhang [1], Christoph Petri [7], and Jianguo Liu [1]

(1 *Key Laboratory of Environmental Optics and Technology, Anhui Institute of Optics and Fine Mechanics, Chinese Academy of Sciences, Hefei 230031, China*)

(2 *Center for Excellence in Regional Atmospheric Environment, Institute of Urban Environment, Chinese Academy of Sciences, Xiamen, 361021, China*)

(3 *University of Science and Technology of China, Hefei, 230026, China*)

(4 *Key Laboratory of Precision Scientific Instrumentation of Anhui Higher Education Institutes, University of Science and Technology of China*)

(5 *Anhui Province Key Laboratory of Polar Environment and Global Change, USTC, Hefei, 230026, China*)

(6 *Department of Atmospheric & Oceanic Sciences, Peking University, Beijing, 100871,China*)

(7 *University of Bremen, Institute of Environmental Physics, P. O. Box 330440, 28334 Bremen, Germany*)

(8 *Royal Belgian Institute for Space Aeronomy (BIRA-IASB), Brussels, Belgium*)

(9 *Department of Physics, University of Toronto, Toronto, Ontario, Canada*)

(10 *Karlsruhe Institute of Technology (KIT), Institute for Meteorology and Climate Research (IMK-ASF), Karlsruhe, Germany*)

(11 *Institute for Space-Earth Environmental Research (ISEE),Nagoya University, Nagoya, 464-8601, Japan*)

(12 *Satellite Observation Center, National Institute for Environmental Studies, Tsukuba, 305-8506, Japan*)

(13 *Institute of Astrophysics and Geophysics, University of Liège, Belgium*)

Correspondence: Cheng Liu (chliu81@ustc.edu.cn) or Lin Zhang (zhanglg@pku.edu.cn)

**Abstract:**

We analyzed seasonality and interannual variability of tropospheric HCN column amounts in densely populated eastern China for the first time. The results were derived from solar absorption spectra recorded with ground-based high spectral resolution Fourier transform infrared (FTIR) spectrometer at Hefei (117°10′E, 31°54′N) between 2015 and 2018. The tropospheric HCN columns over Hefei, China showed significant seasonal variations with three monthly mean peaks throughout the year. The magnitude of the tropospheric HCN column peak in May > September > December. The tropospheric HCN column reached a maximum of $(9.8 \pm 0.78) \times 10^{15}$ molecules/cm$^2$ in May and a minimum of $(7.16 \pm 0.75) \times 10^{15}$ molecules/cm$^2$ in November. In most cases, the tropospheric HCN columns at Hefei (32°N) are higher than the FTIR observations at Ny Alesund (79°N), Kiruna (68°N), Bremen (53°N), Jungfraujoch (47°N), Toronto (44°N), Rikubetsu (43°N), Izana (28°N), Mauna Loa (20°N), La Reunion Maido (21°S), Lauder (45°S), and Arrival Heights (78°S) that are affiliated with the Network for Detection of Atmospheric Composition Change (NDACC). Enhancements of the tropospheric HCN columns were observed between September 2015 and July 2016 compared to the counterpart measurements in other years. The magnitude of the enhancement ranges from 5 to 46% with an average of 22%. Enhancement of tropospheric HCN (ΔHCN) is correlated with the coincident enhancement of tropospheric CO (ΔCO), indicating that enhancements of tropospheric CO and HCN were due to the same sources. The GEOS-Chem tagged CO simulation, the global fire maps and the PSCFs (Potential Source Contribution Function) calculated using back trajectories revealed that the seasonal maxima in May is largely due to the influence of biomass burning in South Eastern Asia (SEAS) $(41 \pm 13.1\%)$, Europe and Boreal Asia



(EUBA) (21 ± 9.3%) and Africa (AF) (22 ± 4.7%). The seasonal maxima in September is largely
due to the influence of biomass burnings in EUBA (38 ± 11.3%), AF (26 ± 6.7%), SEAS (14 ±
3.3%), and Northern America (NA) (13.8 ± 8.4%). For the seasonal maxima in December, dominant
contributions are from AF (36 ± 7.1%), EUBA (21 ± 5.2%), and NA (18.7 ± 5.2%).The tropospheric
HCN enhancement between September 2015 and July 2016 at Hefei (32°N) were attributed to an
elevated influence of biomass burnings in SEAS, EUBA, and Oceania (OCE) in this period.
Particularly, an elevated fire number in OCE in the second half of 2015 dominated the tropospheric
HCN enhancement in September – December 2015. An elevated fire number in SEAS in the first
half of 2016 dominated the tropospheric HCN enhancement in January – July 2016.
**1 Introduction**
Atmospheric hydrogen cyanide (HCN) is an extremely hazardous gas that threaten human
health and terrestrial ecosystems (Andreae and Merlet, 2001; Akagi et al., 2011; Rinsland et al.,
2002). Improved knowledge of the physical and chemical mechanisms which drive the observed
HCN variability is of great significance because HCN plays an important role in the global nitrogen
cycle (Andreae and Merlet, 2001; Li et al., 2003). It is well established that biomass burning is the
major source of tropospheric HCN and industrial emissions contribute additional minor sources of
HCN (Bange and Williams, 2000; Holzinger et al., 1999; Lobert et al., 1990). Li et al. (2009)
estimates a global source of HCN from biomass burning of 0.4 – 3.2 Tg N yr$^{-1}$ and from burning
domestic biofuel of 0.2 Tg N yr$^{-1}$ (Li et al., 2009). Bertschi et al. (2003) estimates a global fossil
fuel combustion source of 0.04 Tg N yr$^{-1}$, negligibly small in comparison (Bertschi et al., 2003).
The principle pathway for HCN sink is ocean uptake which accounts for 0.73 to 1.0 Tg N/year (Li
et al., 2009). Additional minor sinks of HCN are attributed to atmospheric reaction with hydroxyl
radical (OH) and O($^1$D), and photolysis (Li et al., 2000; Nagahama and Suzuki, 2007). The life time
of HCN is 2 – 5 months in the troposphere and several years in the stratosphere. Li et al. (2003,
2009), Lupu et al. (2009), Vigouroux et al. (2012), and Zeng et al. (2012) showed that the observed
variability of HCN can be reproduced by the chemical model simulations where biomass burning
and ocean uptake provide the main source and sink, respectively (Li et al., 2009; Li et al., 2003;
Lupu et al., 2009; Vigouroux et al., 2012; Zeng et al., 2012).
With the rapid economic growth in China over the past three decades, the anthropogenic
emissions have increased dramatically, raising concerns about worsening air quality in China (Tang
et al., 2012; Chan, 2017; Xing et al., 2017; Wang et al., 2017). These emissions are from automobile
exhaust, industrial processes and biomass burning. Many researchers have evaluated regional
emissions in various pollution regions (e.g., the Jing-Jin-Ji region, the Yangtze River Delta region,
and the Pearl River Delta region), but the relative contribution of the biomass burning, automobile
exhaust, and industrial processes is seldom mentioned in the literature (Tang et al., 2012; Chan,
2017; Wang et al., 2017; Sun et al., 2018a; Xing et al., 2017). This is because both industrial
emissions and biomass burning are major sources of the trace gases (e.g. CO, $C_2H_6$ and $CO_2$) that
were used to evaluate regional emissions in the literature, and it is hard to quantify their relative
contribution under the complex pollution condition in China (Chan et al., 2018; Tang et al., 2012;
Wang et al., 2017; Xiaoyan et al., 2010; Xing et al., 2017). It has been proved that HCN is an
unambiguous tracer of biomass burning emission due to its inactive chemical feature and long
lifetime (Rinsland et al., 2002; Zhao et al., 2002). Therefore, measurements of HCN made in
polluted troposphere in eastern China at middle latitude are particularly useful in determining the
potential biomass burning sources that drive the observed tropospheric HCN seasonality and
interannual variability in China.
Ground based high-resolution Fourier Transform Spectroscopy (FTIR) measurements of trace
gases made by Anhui Institute of Optics and Fine Mechanics, Chinese Academy of Sciences
(AIOFM-CAS) at Hefei (117°10′E, 31°54′N, 30 m a.s.l. (above sea level)) is one of few multiyear
time series of trace gases on Asian continent (Sun et al., 2018a; Sun et al., 2018b). These
measurements are crucial to understanding global warming, regional pollution, and long term
transport. Both HCN and CO are regularly measured at Hefei (32°N) in the FTIR observation
routine, influences from biomass burning occurred at a long distance or occurred locally can be
assessed.
In this study, we analyze the first multiyear measurements of tropospheric HCN in densely
populated eastern China. In section 2 the retrieval strategy to derive HCN from high resolution FTIR
spectrometry and the methods for a GEOS-Chem tagged CO simulation and potential source





contribution function (PSCF) calculation are summarized. In section 3 we present the seasonal and
interannual variability of tropospheric HCN columns measured at Hefei (32° N), China and
comparisons with NDACC counterparts. The potential sources that drive the observed HCN
variability are determined by using the GEOS-Chem tagged CO simulation, the global fire maps
and the PSCFs analysis in section 4. The work concludes with a summary in section 5. This study
can improve our understanding of regional biomass burning characteristic and transport, and
contribute to the evaluation of global nitrogen cycle.
**2 Methods**
**2 FTIR observations**
**2.1.1 Site description and instrumentation**
The routine observations of atmospheric trace gases using ground based high-resolution FTIR
spectrometer at Hefei (117°10′E, 31°54′N, 30 m a.s.l.) started in July 2014. Location of Hefei site
alongside those of the NDACC FTIR stations selected for comparison are shown in Fig.1.
Geographical source regions used in the standard GEOS-Chem tagged CO simulation are also
marked in Fig.1. Detailed description of Hefei site can be found in Tian et al., 2017. We follow the
NDACC (Network for Detection of Atmospheric Composition Change, http://www.ndacc.org/, last
accessed on 3 June 2019) requirements, and it is planned to apply for acceptance within the NDACC
in the future.
A Bruker IFS 125 HR with maximum optical path difference (OPD) of 900 cm is used to take
the solar spectra (Tian et al., 2017). Defined as 0.9/OPD, this instrument can reach a highest spectral
resolution of 0.001 cm$^{-1}$. However, all mid-infrared (MIR) spectra are recorded with a spectral
resolution of 0.005 cm$^{-1}$ to ensure a higher signal to noise ratio (SNR) and a faster acquisition time.
This spectral resolution is sufficient to resolve the optical absorption structure of all gases in the
atmosphere. The FTIR spectrometer covered a wide spectral range (about 600 − 4500 cm$^{-1}$) but,
depending on the species, specific detectors and band-pass filters are applied (Sun et al. 2018a). In
this study, the instrument is equipped with a KBr beam splitter & InSb detector & filter no.3 centered
at 2900 cm$^{-1}$ for HCN measurements, and a KBr beam splitter & InSb detector & filter no.4 centered
at 2400 cm$^{-1}$ for CO measurements. The entrance field stop size ranged from 0.80 to 1.5 mm to
adapt the incident radiation. The number of measurements within a day varies from 1 to 20. In total,
there were 651 and 649 days of qualified measurements between 2015 and 2018 for CO and HCN,
respectively.

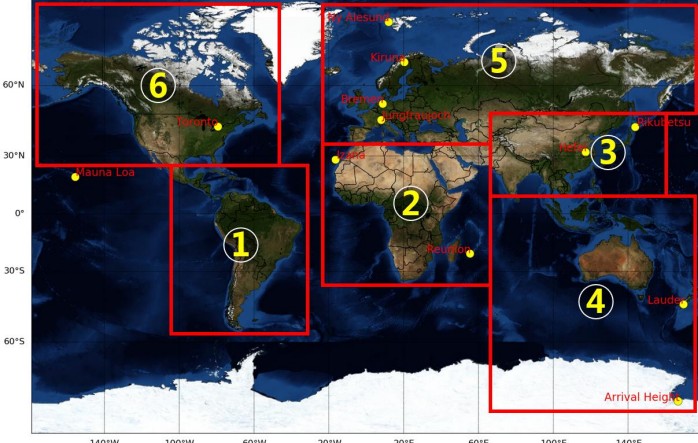

Fig. 1 Location of Hefei site alongside those of the NDACC FTIR stations (yellow dots) that are selected for
comparison. Geographical source regions used in the standard GEOS-Chem tagged CO simulation are also shown.
See Table 3 for latitude and longitude definitions
**2.1.2 Retrieval strategy**
The SFIT4 (version 0.9.4.4) algorithm is used to retrieve the vertical profiles of CO and HCN
(Viatte et al., 2014). Both CO and HCN are standard NDACC species, and we follow the NDACC
recommendation for micro windows (MWs) selection and the interfering gases consideration
(http://www.ndaccdemo.org/, last accessed on 23 May 2019). The retrieval inputs for CO and HCN





are summarized in Table 1. Time series of tropospheric CO columns between 2014 and 2017 at
Hefei (32°N) measured from the FTIR have been reported in Sun et al. (2018a) and the detailed
description of CO profile retrieval can be found therein. Time series of tropospheric HCN columns
at Hefei (32°N) are presented for the first time. Temperature and pressure profiles are extracted
from National Centers for Environmental Protection (NCEP) 6-hourly reanalysis data (De Maziere
et al., 2018) and all spectroscopic absorption parameters are prescribed from HITRAN 2008
database (Rothman et al., 2009). The $H_2O$ *a priori* profile is interpolated from the NCEP 6-hourly
reanalysis data and a priori profiles of other gases are from the WACCM v6 (Whole-Atmosphere
Community Climate Model) special run for NDACC.
Three MWs were used for CO: a strong line at 2057.7–2058 cm$^{-1}$ and two weak lines at
2069.56–2069.76 cm$^{-1}$ and 2157.5–2159.15 cm$^{-1}$ (Sun et al., 2018a). For HCN, two MWs were
used: 3268.00 – 3268.38 cm$^{-1}$ and 3287.00 – 3287.48 cm$^{-1}$ (Mahieu et al., 1997; Lutsch et al., 2016;
Notholt et al., 2000). In order to minimize the cross absorption interference, profiles of $O_3$ and $N_2O$
and columns of $H_2O$, OCS and $CO_2$ are simultaneously retrieved in addition to the CO profile.
Profile of $H_2O$ and columns of $O_3$, $C_2H_2$, and $CH_4$ are simultaneously retrieved in addition to the
HCN profile. No de-weighting SNR is used for HCN and a de-weighting SNR of 500 is used in the
three MWs for CO.
The diagonal elements of *a priori* profile covariance matrices $S_a$ are set to standard deviation
of the WACCM v6 special run for NDACC, and its non-diagonal elements are set to zero. The
diagonal elements of the measurement noise covariance matrices $S_\varepsilon$ are set to the inverse square of
the SNR calculated from each individual spectrum and its non-diagonal elements are set to zero.
The measured instrument line shape (ILS) is included in the retrieval (Hase, 2012; Sun et al., 2018a).

Table 1. Retrieval inputs used for CO and HCN.

| Gases | CO | HCN |
|---|---|---|
| Code | SFIT4 v 0.9.4.4 | SFIT4 v 0.9.4.4 |
| Spectroscopic parameters | HITRAN 2008 | HITRAN 2008 |
| P, T, $H_2O$ profiles | NCEP reanalysis data | NCEP reanalysis data |
| A priori profiles of all gases except $H_2O$ | WACCM v6 | WACCM v6 |
| Micro windows for profile retrievals (cm$^{-1}$) | 2057.7 – 2058 <br> 2069.56–2069.76 <br> 2157.5–2159.15 | 3268.00 – 3268.38 <br> 3287.00–3287.48 |
| Retrieved interfering gases | $O_3$, $N_2O$, $CO_2$, OCS, $H_2O$ | $H_2O$, $O_3$, $C_2H_2$, $CH_4$ |
| SNR for de-weighting | 500 | None |
| $S_a$ | WACCM v6 standard deviation | WACCM v6 standard deviation |
| $S_\varepsilon$ | SNR calculated from each individual spectrum within 2526.23 – 2526.62 | SNR calculated from each individual spectrum within 3381.16 – 3381.54 |
| ILS | LINEFIT145 analysis | LINEFIT145 analysis |
| Error analysis | Systematic error: line intensity, line pressure broadening, line temperature broadening, solar zenith angle, background curvature, solar line strength, optical path difference, field of view, phase | |
| | Random error: <br> -Measurement error <br> -Smoothing error <br> -Interference errors: interfering species, retrieval parameters <br> - Other errors: zero level, temperature | |

**2.1.3 Averaging kernels and error budget**
The partial column averaging kernels of CO and HCN at selected layers are shown in Fig. 2.
The CO averaging kernels have three maxima at the surface, 7 km, and 14 km, respectively. The
HCN averaging kernels have two maxima at 10 km and 16 km, respectively. Both CO and HCN
retrievals show good vertical sensitivity in the whole troposphere where CO exhibits the best
sensitivity with two maxima in the troposphere (Sun et al., 2018a). We can see in Table 2, the typical
degrees of freedom (DOFS) obtained at Hefei (32°N) over the total atmosphere for CO and HCN
are about 2.8 ± 0.3 (1σ) and 1.3 ± 0.2 (1σ), respectively. In this study, only partial columns of CO
and HCN within a broad layer between surface and 15 km are considered. The selected layer
corresponds roughly to the total troposphere over eastern China, as the mean tropopause height
deduced from NCEP reanalysis data is around 15 km over four seasons. The selected layer
corresponds to 2.3 ± 0.2 (1σ) and 1.0 ± 0.1 (1σ) of DOFS for CO and HCN, respectively.

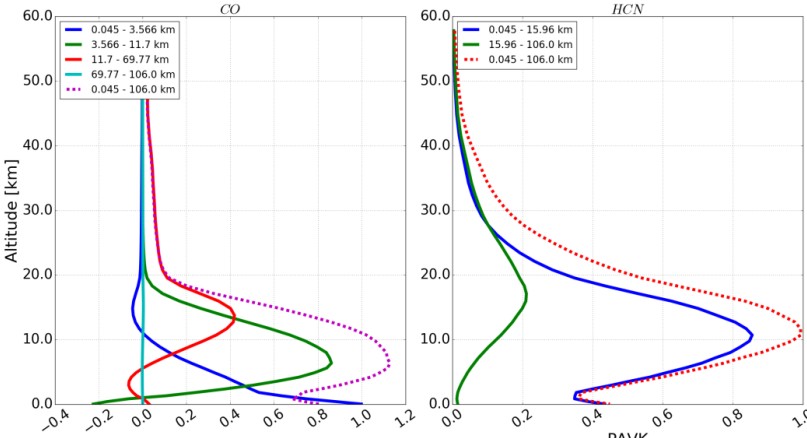

Fig. 2 Partial column averaging kernels (PAVKs) (ppmv / ppmv) for CO and HCN retrievals.

We calculated the error budget following the formalism of (Rodgers, 2000), and separated all error items into systematic error or random error depending on whether they are constant over consecutive measurements, or vary randomly. Table 2 summarizes the random, the systematic, and the combined error budget of tropospheric CO and HCN columns. The error items included in the error budget are listed in Table 1. For CO, the major systematic error is line intensity uncertainty, and the major random error are zero level uncertainty and temperature uncertainty. For HCN, the major systematic error are line intensity uncertainty and line pressure broadening uncertainty, the major random error are smoothing error and measurement error. Total retrieval errors for tropospheric CO and HCN columns between surface and 15 km are estimated to be 8.3 and 14.2%, respectively.

Table 2. Retrieval error budgets and DOFs for tropospheric CO and HCN.

| Gases | CO | HCN |
|---|---|---|
| Temperature uncertainty | 2.5% | 0.2% |
| Zero level uncertainty | 5.2% | 1.5% |
| Retrieval parameters uncertainty | < 0.1% | 2.0% |
| Interfering species uncertainty | < 0.1% | 1.3% |
| Measurement Error | < 0.1% | 6.8% |
| Smooth Error | 0.1% | 11.0% |
| **Total Random Error** | **5.7%** | **13.2%** |
| Background curvature uncertainty | < 0.1% | * |
| Optical path difference uncertainty | < 0.1% | < 0.1% |
| Field of view uncertainty | < 0.1% | < 0.1% |
| Solar line strength uncertainty | < 0.1% | < 0.1% |
| Phase uncertainty | * | < 0.1% |
| Solar zenith angle uncertainty | 0.1% | < 0.1% |
| Line temperature broadening uncertainty | 0.13% | 0.3% |
| Line pressure broadening uncertainty | 0.87% | 3.5% |
| Line intensity uncertainty | 6.0% | 3.7% |
| **Total Systematic Error** | **6.1%** | **5.1%** |
| **Total Errors** | **8.3%** | **14.2%** |
| **DOFS (-)** | **2.2** | **1.0** |

\* Not included into error budget since they are retrieved together with the target gas

**2.2 GEOS-Chem tagged simulation**

To interpret the influence of biomass burning sources on HCN columns at Hefei (32°N), the GEOS-Chem chemical transport model is used (http://geos-chem.org/; Bey et al., 2001b) in a tagged simulation of CO at a horizontal resolution of 2°×2.5° with 47 vertical hybrid levels. GEOS-Chem version 12.2.1 was used and driven by the GEOS-FP assimilated meteorological data observations from the Goddard Earth Observing System (GEOS) of the NASA Global Modeling and Assimilation Office. For driving the GEOS-Chem model, the GEOS-FP meteorological data with a native horizontal resolution of 0.25° latitude × 0.3125° longitude were downgraded to 2° latitude × 2.5° longitude and a vertical resolution of 72 hybrid levels (extending from surface to 0.01 hPa). The





temporal resolution of surface variables and boundary layer height are 1hr and other variables are 3hr.

The GEOS-Chem simulation was initialized with a 1-year spin-up from July 2014 to July 2015. Chemical and transport operator time-steps of 1 hr and 10 min, respectively, were used. Biomass burning emissions are from GFASv1.2 (Global Fire Assimilation System, Kaiser et al., 2012; Giuseppe et al., 2018) which assimilates Moderate Resolution Imaging Spectroradiomter (MODIS) burned area and fire radiative power (FRP) products to estimate emissions for open fires. GFASv1.2 emissions have a $0.1° \times 0.1°$ horizontal resolution with 3-hourly temporal resolution. GFAS was chosen for the availability of emissions over the analysis period from 2015 - 2018. Global anthropogenic and biofuel emissions are from the Community Emissions Data System (CEDS) inventory (Hoesly et al., 2018). In particular, the latest MEIC (the Multi-resolution Emission Inventory for China) inventory is used to provide Chinese anthropogenic emissions (Li et al., 2017). Biogenic emissions of precursor VOCs are from the Model of Emissions of Gases and Aerosols from Nature (MEGANv2.1; Guenther et al., 2012) and biofuel emissions are taken from Yevich and Logan (2003). The main loss mechanism for CO is from photochemical oxidation by the hydroxyl radical (OH). The OH fields are prescribed in the tagged CO simulation and were obtained from the TransCom experiment (Patra et al., 2011) which implements semi-empirically calculated tropospheric OH concentrations from Spivakovsky et al. (2000) to reduce the high bias of OH from the GEOS-Chem full-chemistry simulation (Shindell et al., 2006). Surface emissions in GEOS-Chem are released within the boundary layer, and boundary layer mixing is implemented using the non-local mixing scheme of Holtslag and Boville (1993). Biomass emissions are released by uniformly distributing emissions from the surface to the mean altitude of maximum injection based on the injection height information as described in Rémy et al. (2017) which includes an injection height parameterization by Sofiev et al. (2012) and a plume rise model by Freitas et al. (2007).

GEOS-Chem version 12.2.1 tagged CO simulation includes the improved secondary CO production scheme of Fisher et al. (2017), which assumes production rates of CO from $CH_4$ and NMVOC (non-methane volatile organic compounds) oxidation from a GEOS-Chem full-chemistry simulation therefore reducing the mismatch between the CO-only simulation and the full-chemistry simulation.

The tracers of anthropogenic, biomass burning, $CH_4$ and NMVOC oxidations are implemented following the standard GEOS-Chem tagged CO simulation (Giglio et al., 2013). In this study, we don't investigate the influence of each individual anthropogenic and oxidation source tracer. For investigation of the influence of biomass burning sources, the regional definitions of all biomass burning tracers are shown in Fig. 1 and tabulated in Table 3.

Table 3. Regional definitions of all biomass burning tracers implemented in the standard GEOS-Chem tagged CO simulation

| No. | Tracer | Description | Region |
|---|---|---|---|
| 1 | SA | Biomass burning CO emitted over South America | 112.5°W - 32.5°W; 56°S - 24°N |
| 2 | AF | Biomass burning CO emitted over Africa | 17.5°W -70.0°E; 48.0°S - 36.0°N |
| 3 | SEAS | Biomass burning CO emitted over Southeast Asia | 70.0°E - 152.5°E; 8.0°N - 45.0°N |
| 4 | OCE | Biomass burning CO emitted over Oceania | 70.0°E - 170.0°E; 90.0°S - 8.0°N |
| 5 | EUBA | Biomass burning CO emitted over Europe and Boreal Asia | 17.5°W - 72.5°E; 36.0°N - 45.0°N and 17.5°W - 172.5°E; 45.0°N - 88.0°N |
| 6 | NA | Biomass burning CO emitted over North America | 173°W - 50°W; 24.0°N - 88.0°N |

**2.3 Potential source contribution function**

The potential source contribution function (PSCF) assumes that back trajectories arriving at times of higher concentrations likely point to the more significant pollution directions (Ashbaugh et al., 1985). PSCF has been applied in many studies to locate air masses associated with high levels of air pollutants (Kaiser et al., 2007; Dimitriou and Kassomenos, 2015; Yin et al., 2017). In this study, PSCF values were calculated using back trajectories that were calculated by HYSPLIT. The top of the model was set to 10 km. The PSCF values for the grid cells in the study domain were based on a count of the trajectory segment that terminated within each cell (Ashbaugh et al., 1985).





The number of endpoints that fall in the $ij$th cell is designated $n_{ij}$. The number of endpoints for the
same cell having arrival times at the sampling site corresponding to concentrations higher than an
arbitrarily set criterion is defined to be $m_{ij}$. In this study, we calculated the PSCF values based on
trajectories corresponding to concentrations that exceeded the monthly mean level of tropospheric
HCN column during measurement. The PSCF value for the $ij$th cell is then defined as:

$$PSCF_{ij} = m_{ij}/n_{ij} \qquad (1)$$

The PSCF value can be interpreted as the conditional probability that the concentrations of a
given analyte greater than the criterion level are related to the passage of air parcels through the $ij$th
cell during transport to the receptor site. That is, cells with high PSCF values are associated with
the arrival of air parcels at the receptor site that have concentrations of the analyte higher than the
criterion value. These cells are indicative of areas of 'high potential' contributions for the constituent.
Identical $PSCF_{ij}$ values can be obtained from cells with very different counts of back-trajectory
points (e.g., grid cell A with $m_{ij} = 400$ and $n_{ij} = 800$ and grid cell B with $m_{ij} = 4$ and $n_{ij} = 8$). In this
extreme situation grid cell A has 100 times more air parcels passing through than grid cell B.
Because of the sparse particle count in grid cell B, the PSCF values are more uncertain. To account
for the uncertainty due to low values of $n_{ij}$, the PSCF values were scaled by a weighting function
$W_{ij}$ (Polissar et al., 1999). The weighting function reduced the PSCF values when the total number
of endpoints in a cell was less than approximately 3 times the average value of the end points per
cell. In this case, $W_{ij}$ was set as follows:

$$W_{ij} = \begin{cases} 1.00 & n_{ij} > 3N_{ave} \\ 0.70 & 3N_{ave} > n_{ij} > 1.5N_{ave} \\ 0.42 & 1.5\,N_{ave} > n_{ij} > N_{ave} \\ 0.05 & N_{ave} > n_{ij} \end{cases} \qquad (2)$$

where $Nave$ represents the mean $n_{ij}$ of all grid cells. The weighted PSCF values were
obtained by multiplying the original PSCF values by the weighting factor.
**3 FTIR time series and comparisons with NDACC counterparts**
The new HCN data are compared with the counterparts regularly measured at eleven NDACC
stations to investigate the representativeness of the observation site at Hefei (32°N) in polluted
eastern China. These NDACC stations cover over a wide latitude range from 77.8°S to 78.9°N and
a wide longitude range from 79°W to 170°E (http://www.ndaccdemo.org/, last access on 19 July
2019). Most of these NDACC stations use the same instrument and retrieval algorithm as those of
Hefei (32°N). Alternatively, the high resolution spectrometers Bruker 125M, 120HR, or Bomem
DA8 and the retrieval algorithm PROFFIT are used in other stations. It has been demonstrated that
the profiles derived from these different instruments and algorithms are in excellent agreement
(Hase et al., 2004; De Maziere et al., 2018). In addition, we show the time series of tropospheric
CO columns, also measured with FTIR spectrometer, because we will discuss the correlation
between HCN and CO, and quantify the influence of biomass burning sources on HCN columns at
Hefei (32°N) by using a tagged CO simulation. The upper limit of 15 km is above the tropopause at
most of the NDACC stations. For most NDACC stations, the surface – 15 km layer is a mixture of
the total troposphere and a part of stratosphere. However, we did not find major changes in the
results of this study when choosing a lower upper limit such as 12 km. Thus we have chosen the
same upper limits for all stations. The geolocations of all FTIR stations and their seasonal maximum,
minimum and variabilities are summarized in Table 4.
Table 4. Tropospheric HCN and CO columns at Hefei (32°N), China from 2015 to 2018 alongside those of the
NDACC FTIR stations. All stations are organised as a function of decreasing latitude.

| Station | Location (Lon., Lat., Alt. in km) | Instrument | Algorithm | Maximum (molecules cm$^{-2}$) | | Minimum (molecules cm$^{-2}$) | |
|---|---|---|---|---|---|---|---|
| | | | | HCN ($10^{15}$) | CO ($10^{18}$) | HCN ($10^{15}$) | CO ($10^{18}$) |
| Ny Alesund | (12°E, 79°N, 0.02) | 125HR | SFIT4 | 5.94± 1.20 (August) | 2.11 ± 0.11 (March) | 3.75 ± 0.37 (March) | 1.56 ± 0.12 (July) |
| Kiruna | (20°E, 68°N, 0.42) | 125HR | PROFFIT | 5.81 ± 0.58 (August) | 2.1 ± 0.01 (January) | 2.43 ± 0.27 (January) | 1.45 ± 0.09 (July) |
| Bremen | (9°E, 53°N, 0.03) | 125HR | SFIT4 | 6.11 ± 0.87 (August) | 2.32 ± 0.13 (March) | 2.85 ± 0.25 (July) | 1.63 ± 0.19 (July) |
| Jungfraujoch | (8°E, 46.5°N, 3.58) | 125HR | SFIT4 | 4.68 ± 0.63 (May) | 1.14 ± 0.08 (March) | 2.1 ± 0.29 (February) | 0.88 ± 0.08 (July) |
| Toronto | (79°W, 44°N, 0.17) | Bomem DA8 | SFIT4 | 5.92 ± 1.13 (May) | 2.19 ± 0.15 (April) | 3.12 ± 1.02 (November) | 1.74 ± 0.1 (October) |



| Rikubetsu | (144°E, 43°N, 0.38) | 125HR | SFIT4 | 7.0 ± 1.92 (May) | 2.32 ± 0.31 (March) | 2.86 ± 0.44 (February) | 1.79 ± 0.14 (October) |
|---|---|---|---|---|---|---|---|
| **Hefei** | **(117°E, 32°N, 0.03)** | **125HR** | **SFIT4** | **9.8 ± 0.78 (May)** | **3.38 ± 0.43 (February)** | **7.16 ± 0.75 (November)** | **2.29 ± 0.48 (July)** |
| Izana | (16°W, 28°N, 2.37) | 125HR | PROFFIT | 5.33 ± 1.2 (May) | 1.41 ± 0.14 (April) | 2.59 ± 0.28 (October) | 1.1 ± 0.08 (October) |
| Mauna Loa | (24°W, 20°N, 3.40) | 125M | SFIT4 | 4.49 ± 1.8 (April) | 1.36 ± 0.31 (April) | 2.07 ± 0.43 (August) | 0.8 ± 0.04 (August) |
| La Reunion Maido | (55°E, 21°S, 2.16) | 125HR | SFIT4 | 6.91 ± 2.45 (November) | 1.46 ± 0.17 (October) | 2.56 ± 0.48 (May) | 1.0 ± 0.1 (April) |
| Lauder | (170°E, 45°S, 0.37) | 120HR | SFIT4 | 5.29 ± 1.18 (November) | 1.28 ± 0.19 (October) | 1.94 ± 0.28 (July) | 0.89 ± 0.09 (February) |
| Arrival Heights | (167°E, 78°S, 0.2) | 120HR | SFIT4 | 3.22 ± 0.51 (February) | 1.0 ± 0.04 (October) | 1.78 ± 0.21 (September) | 0.67 ± 0.03 (April) |

**3.1 Seasonal variation**
The monthly means of the tropospheric CO and HCN columns at the twelve FTIR stations are
shown in Fig. 3. As commonly observed at Hefei (32°N), three monthly mean peaks are evident for
tropospheric HCN and CO columns. The magnitude of the tropospheric HCN peak at Hefei (32°N)
in May > September > December. While for tropospheric CO column, the magnitude of the peak at
Hefei (32°N) in February > September > December. For tropospheric HCN and CO columns, the
timing of the monthly mean maximum and minimum are different, but the timing of the smaller two
monthly mean peaks are the same. The tropospheric CO and HCN columns at Hefei (32°N) show
similar seasonal variability throughout the year except March to May, when the variability is
opposite. The biggest contrast in terms of seasonal cycle occurs in May.

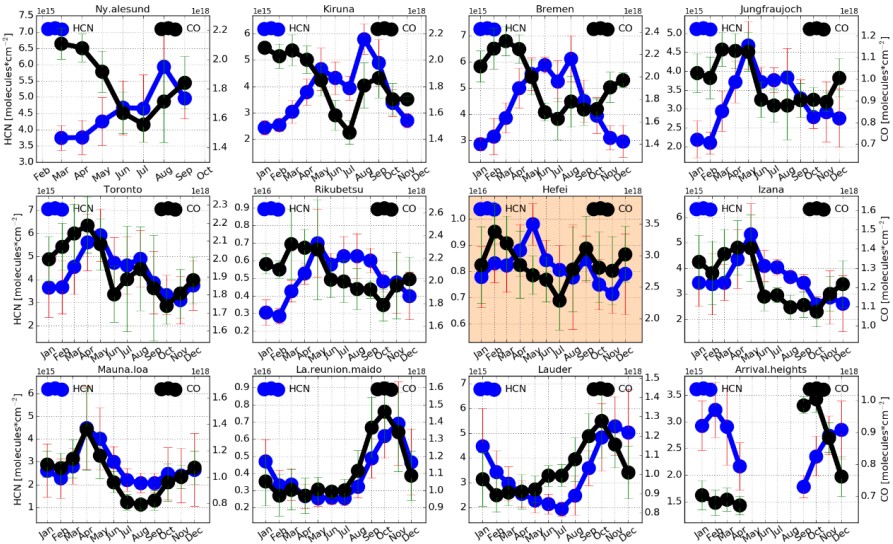

Fig. 3. Monthly means of the tropospheric CO and HCN columns at Ny Alesund, Kiruna, Bremen, Jungfraufoch,
Toronto, Rikubetsu, Hefei, Izana, Mauna Loa, La Reunion Maido, Lauder, and Arrival Heights from 2015 to 2018.
Vertical error bars represent 1σ within that month. All stations are organised as a function of decreasing latitude.
The tropospheric HCN and CO columns at Hefei (32°N) are higher than the NDACC FTIR
observations (see Fig. A1). The tropospheric HCN column reached a maximum of (9.8 ± 0.78) ×
$10^{15}$ molecules/cm² in May and a minimum of (7.16 ± 0.75) × $10^{15}$ molecules/cm² in November.
The tropospheric CO column reached a maximum of (3.38 ± 0.43) × $10^{18}$ molecules/cm² in February
and a minimum of (2.29 ± 0.48) × $10^{18}$ molecules/cm² in July (Table 4). In comparison, the seasonal
maxima and minima of tropospheric HCN columns at the selected NDACC FTIR stations varied
over (3.22 ± 0.51) to (7.0 ± 1.92) × $10^{15}$ molecules/cm² and (1.78 ± 0.21) to (3.75 ± 0.37) × $10^{15}$
molecules/cm², respectively. The seasonal maxima and minima of tropospheric CO columns at the
selected NDACC FTIR stations varied over (1.0 ± 0.04) to (2.32 ± 0.31) × $10^{18}$ molecules/cm² and
(0.67 ± 0.03) to (1.79 ± 0.14) × $10^{18}$ molecules/cm², respectively (Table 4).
In the northern hemisphere, the timing of the seasonal maxima for tropospheric HCN columns



generally occur in spring or summer, and for CO occur in winter or spring. While in the southern
hemisphere, the timing of the seasonal maxima for both tropospheric HCN and CO columns occur
in autumn or winter.
**3.2 Interannual variability and enhancement**

5       In order to study the interannual variability of HCN and CO, fractional differences in the
tropospheric HCN and CO columns relative to their seasonal mean values represented by the cosine
fitting at the twelve FTIR stations are shown in Fig.4 and Fig.5, respectively. Enhancements of both
tropospheric HCN and CO columns between September 2015 and July 2016 at Hefei (32°N) were
observed compared to the measurements in other years. For HCN, the magnitude of the
enhancement ranges from 5 to 46% with an average of 26%. The significant enhancements occurred
in December 2015 and May 2016 with peaks of 46% and 38%, respectively. By contrast, the
magnitude of the enhancement in tropospheric CO column at Hefei (32°N) between September 2015
and July 2016 ranges from 4 to 59% with an average of 27%.The tropospheric CO columns were
elevated over its seasonal means by more than 20% from March to April 2016. In addition, an
enhancement magnitude of more than 40% were occasionally observed in August and September
for both HCN and CO at Hefei (32°N).
The enhancements of both tropospheric HCN and CO columns within the same period were
also observed at the selected NDACC stations except Ny Alesund (79°N) and Kiruna (68°N). The
winter enhancements were not shown over Ny Alesund (79°N) and Kiruna (68°N) because of the
polar night in the Arctic which interrupted the observations in winter. The magnitude of the
enhancement in tropospheric HCN column at the selected NDACC stations between September
2015 and July 2016 ranges from 3 to 213%, and for CO ranges from 4 to 62%.

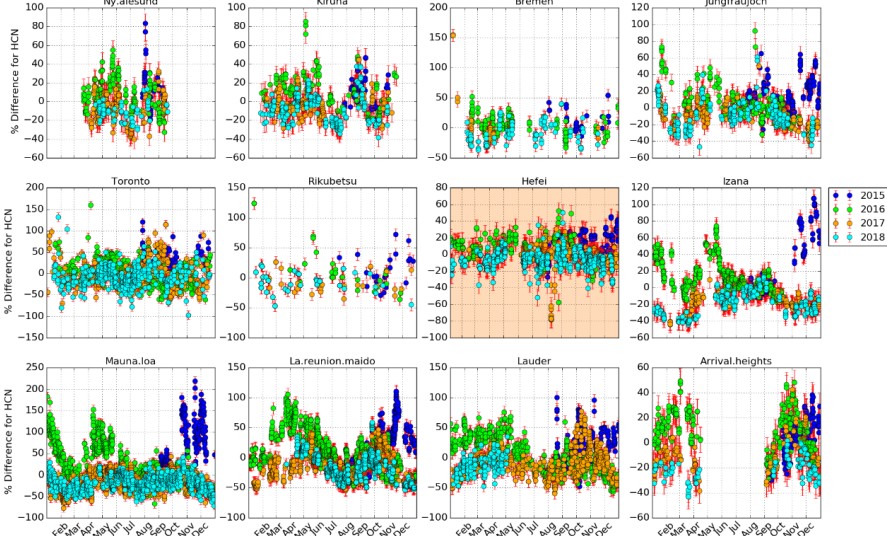


Fig.4. Fractional difference in the partial columns (surface - 15 km) of HCN from 2015 to 2018 at Ny Alesund,
Kiruna, Bremen, Jungfraufoch, Toronto, Rikubetsu, Hefei, Izana, Mauna Loa, La Reunion Maido, Lauder, and
Arrival Heights relative to their seasonal mean values. Vertical error bars represent the estimated retrieval errors.
All stations are organised as a function of decreasing latitude.

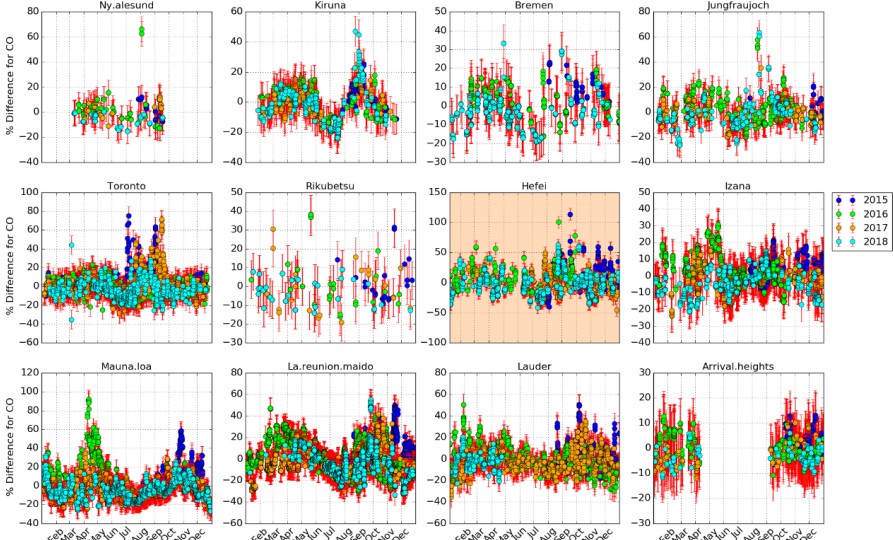

Fig.5. The same as Fig.4 but for CO.
**3.3 Correlation with CO and enhancement ratios**
The tropospheric HCN columns at the twelve FTIR stations have been plotted against the
coincident CO partial columns (Fig.6). In Fig.7, the correlations between the tropospheric HCN and
CO columns at Hefei (32°N) for all spectra recorded throughout the year (gray dots) and those
recorded within the selected periods (green dots) are compared. We followed the least squares
procedure of York et al., 2004 to proceed a linear regression for the coincident measurements, and
incorporated the errors in both ordinal and abscissa coordinates into the uncertainty estimation.
Since different atmospheric chemistry processes control the abundance of CO and HCN,
moderate overall correlations between HCN and CO tropospheric columns were present at
Jungfraujoch (47°N) and Rikubetsu (43°N), and negative overall correlations were present at Ny
Alesund (79°N), Kiruna (68°N), Bremen (53°N), and Arrival Heights (78°S). However, high
correlation of these two species were seen at Toronto (44°N), Hefei (32°N), Izana (28°N), Mauna
Loa (20°N), La Reunion Maido (21°S), and Lauder (45°S) throughout the year probably because
the portion of the fire-affected seasonal measurements at these stations are larger than those at other
stations (Fig.6). For the measurements at Hefei (32°N), the high correlations between HCN and CO
tropospheric columns deduced from the measurements without March and April ($R$=0.67, Fig.7 (a)),
in May ($R$=0.69, Fig.7 (b)), in September($R$=0.77, Fig.7 (c)), and in December ($R$=0.65, Fig.7 (d))
are consistent with that deduced from all measurements ($R$=0.70) (Table 5). However, the
correlation slope for the May, September, and December tropospheric columns differ from the
annual one, indicating different biomass burning sources in different periods.
For fire-affected measurements, the slope $\Delta HCN/\Delta CO$ defined as enhancement ratio (EnhR$_{HCN}$)
is an important parameter in quantification of biomass burning emissions (Holzinger et al., 1999;
Lutsch et al., 2016; Rinsland et al., 2002; Viatte et al., 2015; Vigouroux et al., 2012; Zhao et al.,
2000). Depending on the burnt biomaterials, fire type, the phase of the fire, and the travel time of
the plumes, the reported EnhR$_{HCN}$ varied by 2 orders of magnitude. The mean EnhR$_{HCN}$ of $1.34\times10^{-3}$
at Hefei (32°N) falls between the wide range of the HCN/CO ratios measured in laboratory ($0.4 -$
$7.1\times10^{-3}$ in the work of (Yokelson et al., 1997) and $0.4 - 2.6\times10^{-3}$ in the work of (Holzinger et al.,
1999), and $0.94 - 7.4\times10^{-3}$ in the NDACC FTIR measurement counterparts (Fig. 6). The mean
EnhR$_{HCN}$ at Hefei (32°N) is close to that at Rikubetsu (43°N) indicates these two Asian stations
share similar biomass burning sources through the year. The mean EnhR$_{HCN}$ at Hefei (32°N) is lower
than those measured at Jungfraujoch (47°N), Toronto (44°N), Izana (28°N), Mauna Loa (20°N),
Lauder (45°S), and La Reunion Maido (21°S) because the emissions of crop residue burning which
dominates the HCN enhancements at Hefei (32°N) is lower than those of the boreal or tropical forest
burning, which account for the HCN enhancements at aforementioned NDACC stations (Akagi et
al., 2011; Akagi et al., 2012; Rinsland et al., 2007; Vigouroux et al., 2012). On the other hand, the





Hefei (32°N) site located in the densely populated part of China, emissions of fossil fuel combustion
such as automobile exhaust and industrial processes could elevate the CO background level and
hence lessen the EnhR$_{HCN}$.

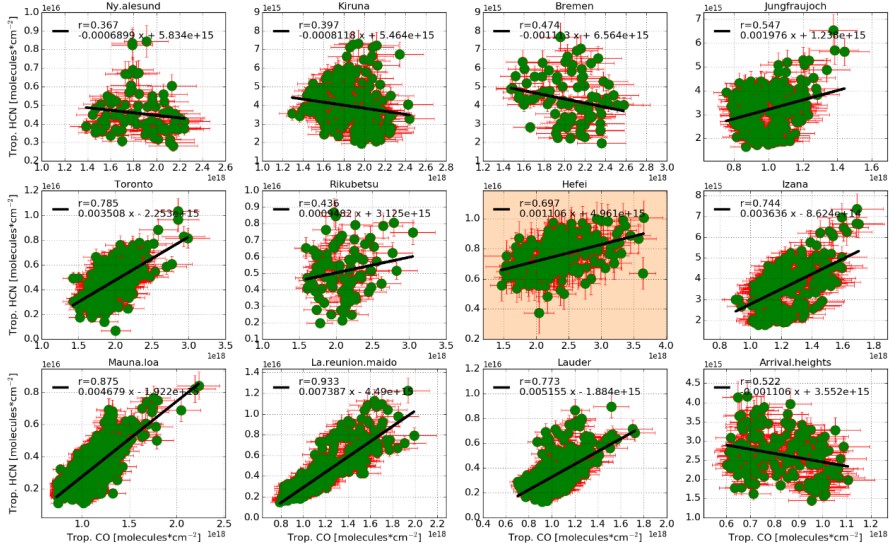

Fig. 6. Correlation plots of daily mean partial columns (surface - 15 km) of HCN versus CO (molecules/cm$^2$). The
linear equation of the fit and the resulting correlation coefficient $r$ are shown. The black line is a linear least-squares
fit of respective data. All stations are organised as a function of decreasing latitude. Error bars represent the retrieval
uncertainties.
Table 5. Correlation between HCN and CO tropospheric columns within each selected period at Hefei (32°N),
China. $N$ is the number of points, $R$ is the correlation coefficient and EnhR$_{HCN}$ is the enhancement ratio.

| Gas | Period | without March and April | May | September | December | Mean |
|---|---|---|---|---|---|---|
| HCN | $N$ | 239 | 26 | 56 | 35 | - |
| | $R$ | 0.67 | 0.69 | 0.77 | 0.65 | 0.7 |
| | EnhR×10$^{-3}$ | 1.06 | 1.48 | 1.29 | 1.52 | 1.34 |

Fig. 7. Correlation plots of daily mean tropospheric columns of HCN versus CO (molecules/cm$^2$) at Hefei (32°N).



The gray dots represent all measurements and the green dots represent the measurements within the selected period: (a) measurements without March and April; (b) measurements in May; (c) measurements in September; (d) measurements in December. The linear equation of the fit and the resulting correlation coefficient $r$ are shown. The black line is a linear least-squares fit of the gray data and the blue line is for the green data. Error bars represent the retrieval uncertainties.

## 4 Source attribution

In order to determine what drives the seasonality and interannual variability of tropospheric HCN in eastern China, it is necessary to match the observed time series with actual biomass burning events, and show that the generated plumes are capable of travelling to the observation site. We did this by using various independent data sets.

1. The 1-hourly instantaneous CO VMR (volume mixing ratio) profiles of the tracers listed in Table 3 provided by a GEOS-Chem tagged CO simulation performed as described in Section 2.2.

2. The global fire atlas data archived by the Fire Information for Resource Management System (FIRMS) which generates fire information from NASA's Moderate Resolution Imaging Spectroradiometer (MODIS) and NASA's Visible Infrared Imaging Radiometer Suite (VIIRS) (https://firms.modaps.eosdis.nasa.gov/download/, last access on 23 May 2019). We have only taken the fire number with a retrieval confidence value of larger than 60% into account.

3. Three dimensional kinematic back trajectories at designated elevations calculated by the Air Resources Laboratory (ARL, http://ready.arl.noaa.gov/HYSPLIT.php, last accessed on 23 May 2019) Hybrid Single Particle Lagrangian Integrated Trajectory (HYSPLIT) model using Global Data Assimilation System (GDAS) meteorological fields (https://ready.arl.noaa.gov/gdas1.php, last accessed on 23 May 2019).

4. The PSCF values calculated by MeteoInfo as described in Section 2.3 using HYSPLIT back trajectories (http://meteothink.org/index.html, last accessed on 17 December 2019).

### 4.1 Attribution for the seasonality

The GEOS-Chem tagged CO simulation provides a means of evaluating the contribution of CO from anthropogenic, biomass burning and oxidation sources to the measured CO columns at Hefei (32°N). Source attribution is performed as follows. First, the GEOS-Chem CO VMR profiles of all tracers in the grid box containing the Hefei (32°N) site were converted to partial column profiles and linearly interpolated and regridded onto the FTIR vertical retrieval grid. This was necessary in order to account for the differences in the vertical levels of the model and the FTIR (Barret et al., 2003). Then, The GEOS-Chem CO partial column profiles are smoothed by the normalized FTIR CO total column averaging kernel following Rodgers and Connor (2003). The GEOS-Chem CO profiles, FTIR CO profiles and total column averaging kernels are daily averaged and the daily averaged GEOS-Chem profiles are subsequently smoothed. Fig.8 shows the daily-averaged GEOS-Chem and FTIR CO tropospheric columns (surface-15 km) for the simulation period from 2015 - 2018. The relative contribution of anthropogenic, biomass burning and oxidation tracers are also shown. The GEOS-Chem and FTIR CO tropospheric columns are in good agreement.

The combination of the anthropogenic source and the oxidations of $CH_4$ and NMVOCs is the greatest contribution to the tropospheric CO column at Hefei (32°N). The magnitude of this combination source varies over 80 to 95%. In contrast, the magnitude of biomass burning source varies over 5 to 20%. As shown in Fig.9, the anthropogenic, biomass burning and oxidation sources are all seasonal dependent due to the magnitude of the emissions and the influence of seasonally variable transport. The onset of the anthropogenic contribution begins in July with a maximum in December. In contrast to the anthropogenic influence, the onset of the oxidation contribution begins in January with a maximum in July, as a result of maximum NMVOC emissions in Summer (Sun et al., 2018b). For biomass burning contribution, two onsets were observed. One begins in January with a maximum in April and the other one begins in July with a maximum in October.

After normalizing each biomass burning tracer listed in Table 3 to the total biomass burning contribution, the normalized relative contribution of each individual biomass burning tracer to the total biomass burning associated CO tropospheric column was obtained in Fig.10. The results show that the seasonal maxima in May is largely due to the influence of SEAS biomass burning (41 ± 13.1%). Moderate contributions from EUBA (21 ± 9.3%) and AF (22 ± 4.7%), and small contributions from SA (7.8 ± 2.9%), OCE (1.5 ± 0.8%), and NA (7.7 ± 1.9%) are also observed. The seasonal maxima in September is largely due to the influence of EUBA (38 ± 11.3%) and AF (26 ± 6.7%) biomass burnings. Remaining contributions are from SA (5.1 ± 2.7%), SEAS (14 ± 3.3%),





OCE (8.9 ± 7.4%), and NA (13.8 ± 8.4%). For the seasonal maxima in December, contributions
from AF, SA, SEAS, EUBA, OCE, and NA are 36 ± 7.1%, 11 ± 1.9%, 11 ± 3.6%, 21 ± 5.2%, 4.8 ±
2.7%, and 18.7 ± 5.2%, respectively.
Fig. 8. Daily-mean CO tropospheric column time series of FTIR and GEOS-Chem (top panel) from 2015-2018 at
Hefei (32°N). The bottom panel shows the relative contribution (%) of the anthropogenic, biomass burning, and
oxidation tracers in the GEOS-Chem simulation to the total CO tropospheric column.

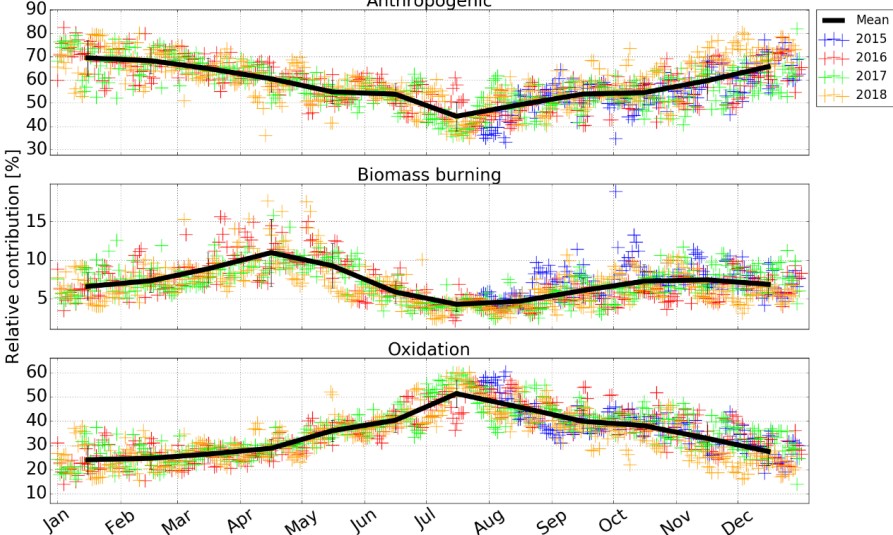

Fig. 9. Seasonality of the relative contribution (%) of the anthropogenic, biomass burning, and oxidation tracers in
the GEOS-Chem simulation to the total CO tropospheric column.

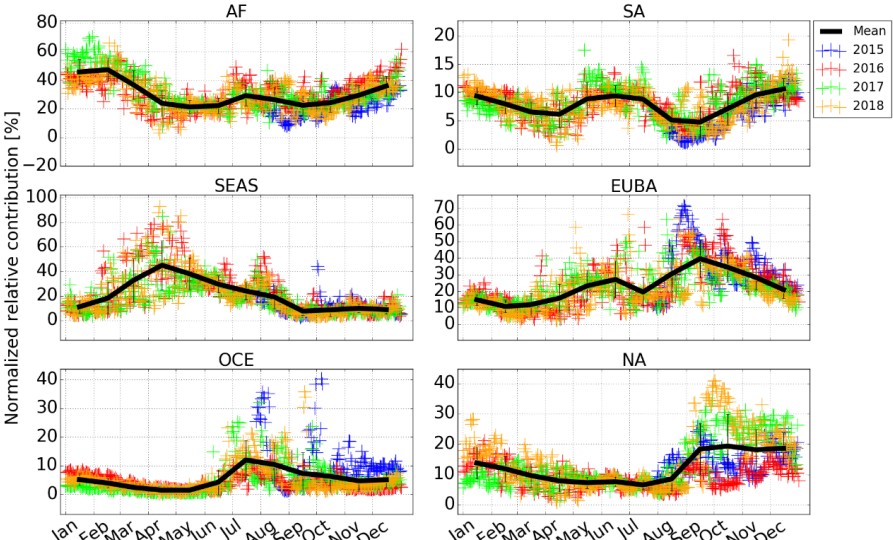

Fig. 10. Seasonality of the normalized relative contribution (%) of the AF, SA, SEAS, EUBA, OCE, and NA biomass burning tracers in the GEOS-Chem simulation to the total biomass burning associated CO tropospheric column.

**4.2 Attribution for transport pathway**

For each seasonal enhancement of the tropospheric HCN, transport pathway is determined as follows. First, the GEOS-Chem tagged CO simulation is used to calculate the relative contribution of each biomass burning tracer (Fig. 10). For the tracer with a high contribution, the FIRMS global fire map is used to search for potential fire events occurred before the timing of tropospheric HCN enhancement within one month period. Then, we generated an ensemble of HYSPLIT back trajectories with different travel times and arrival altitudes to judge whether these plumes are capable of travelling to the observation site. For example, for each intensive biomass burning event detected at a specific period, we generated ten back trajectories at different arrival altitudes ranging from 1.5 to 12 km, and modified the end time of these back-trajectories within one day of the observed enhancement. If the back-trajectories intersect a region where the FIRMS fire data indicates an intensive fire event and the travel duration is within a reasonable range, then this specific fire event could contribute to the observed enhancements at Hefei (32°N) in eastern China. The transport pathway for this enhancement is finally determined.

Fig. 11 demonstrates travel trajectories of the plumes occurred in AF, SEAS & OCE, EUBA, and NA that reached Hefei (32°N) through long range transport. Fig. 12 shows the PSCFs calculated using 13-day HYSPLIT back trajectories that are coincident with the FTIR measurement time. The eastern China, South Asia, Central Asia, Eastern Europe, and Northern Africa had high PSCF weight values in both the first and second half year. The large areas of Southeastern Asian countries including Philippines, Malaysia, and Indonesia, and the Eastern North America were the additional regions with potentially high PSCF weight values in the second half year. Generally, trajectories with the same travel time in the second half year are longer than those in the first half year, resulting in broader areas with potentially high PSCF weight values.

As Figs.13 and 14 shown, the seasonal biomass burning typically occurs in July – September in southern Africa and in November – February in central Africa. These AF emissions can be transported to eastern China along with the southwestern wind which contributed 25 – 45% of the tropospheric HCN in these periods. The seasonal biomass burning typically occurs in March – May and July – November in central Europe, and in June – September in Siberia. These EUBA emissions can be transported to eastern China along with the northwestern or northern wind which contributed 27 – 40% of the tropospheric HCN in these periods. The seasonal biomass burning typically occurs in March – May in India and South Asia peninsula. Drives by the Asian monsoon anticyclone (AMA), the dominant circulation feature in the Indian–Asian upper troposphere–lower stratosphere region during the Asian monsoon, these emissions can be transported to eastern China which contributed to the tropospheric HCN peak in May. The seasonal biomass burning typically occurs

in March – May, July – September, and November – December in the eastern part of China. All
these emissions can be transported to the observation site at Hefei (32°N) under favorable
meteorological condition and thus contribute to all the seasonal tropospheric HCN peaks. The SEAS
contribution (mainly China, India and South Asia peninsula) varies over 25 to 80% in March to
August.
Additionally, a small to moderate portion of wildfire events in central SA, eastern NA, and
Northern OCE in autumn or winter could transport to the observation site through long distance
atmospheric circulation, which contributed 5 – 20% of the tropospheric HCN in these periods.

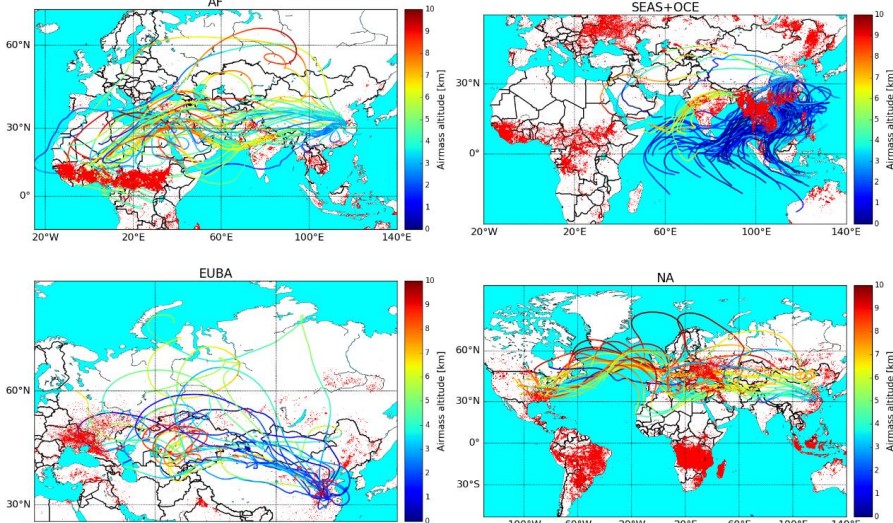

Fig.11. Travel trajectories of the plumes occurred in AF, SEAS & OCE, EUBA, and NA that reached Hefei (32°N)
through long range transport. Travel times are 13, 7, 10, and 14 days, respectively. For clarity, only few trajectories
are selected for demonstration.

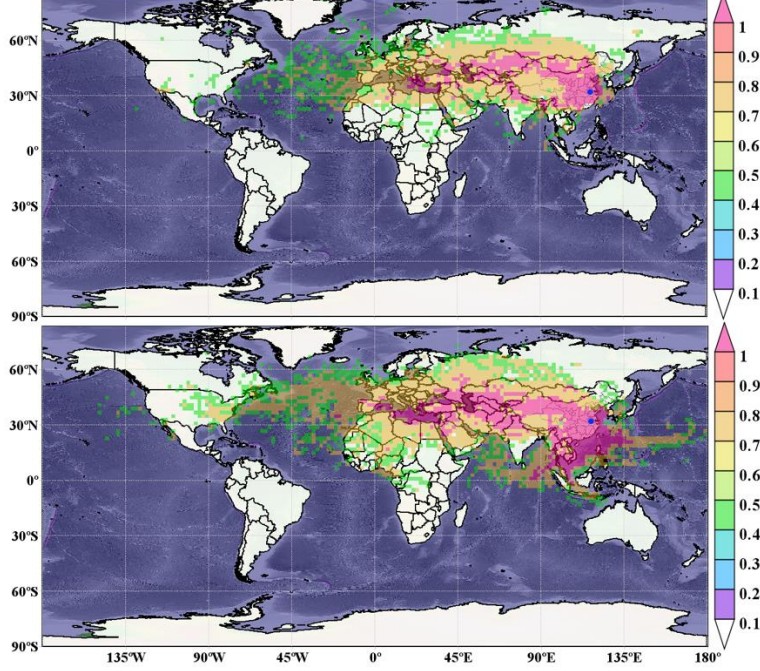



Fig.12. Likely source areas of air mass associated with higher HCN concentrations at Hefei (32°N) in the first half
year (top panel) and the second half year (bottom panel) identified using PSCF.

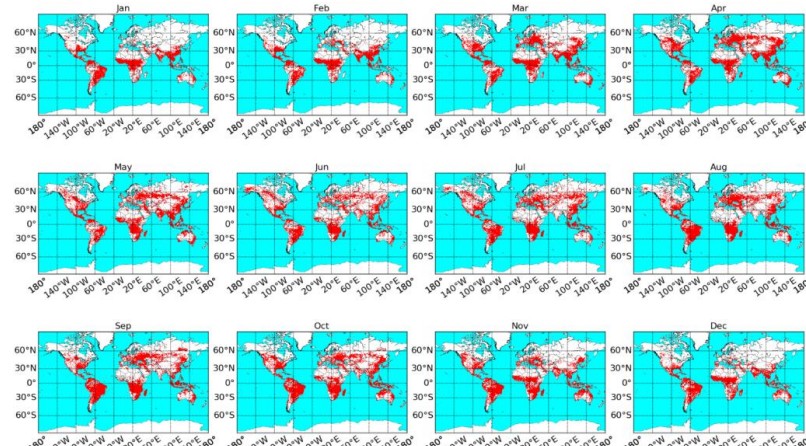

Fig. 13. Global fire map in January to December 2015 accumulated from the FIRMS fire atlas.

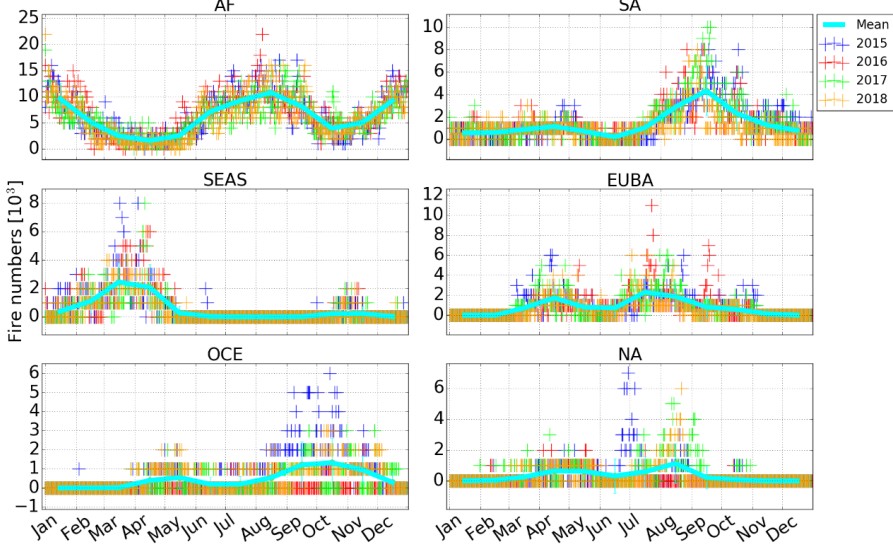

Fig.14. Seasonality of total fire numbers within the AF, SA, SEAS, EUBA, OCE, and NA tracers. All data are
accumulated from the FIRMS fire atlas.
**4.3 Attribution for interannual variability**

9        In Fig. 9, the biomass burning contribution was elevated by $5-15\%$ between September 2015
and July 2016, while no elevations were observed for anthropogenic and oxidation influence. As a
result, enhancements of both tropospheric HCN and CO columns between September 2015 and July
2016 at Hefei (32°N) were attributed to an elevated influence of biomass burning. In Fig.10, the
relative contribution (%) of the SEAS, EUBA, and OCE biomass burning tracers to the total biomass
burning associated CO tropospheric column were elevated by $5-20\%$, $8-27\%$, $8-31\%$,
respectively, in the second half of 2015 compared to the same period in other years. The relative
contribution (%) of the SEAS and OCE biomass burning tracers to the total biomass burning
associated CO tropospheric column were elevated by $8-39\%$ and $2-7\%$, respectively, in the first
half of 2016 compared to the same period in other years.

19       The statistical results of the FIRMS fire atlas data in Fig.14 show that, the fire numbers in the
SEAS, EUBA, and OCE regions elevated by 21.89%, 15.72%, and 32.68% between September


2015 and July 2016 compared to the same period in other years. These elevated fire numbers in
EUBA, SEAS and OCE driven the enhancements of tropospheric HCN and CO columns between
September 2015 and July 2016 at Hefei (32°N). Particularly, the number of fires in OCE in the
second half of 2015 was greatly elevated in comparison with the other years, acting as a dominant
source of tropospheric HCN enhancement in September – December 2015. The fire numbers
elevated significantly in the SEAS region in the first half of 2016, which dominated the tropospheric
HCN enhancement in January – July 2016.
Compared to the northwestern part of China such as the Xinjiang province and the Tibet plateau,
the densely populated eastern parts of China are more suitable for crop planting because of fertile
soil and adequate water resources. Historically, Chinese farmers burned their crop residue (such as
rice, corn, and wheat straws) after harvest to fertilize the soil for the coming farming season. Post-
harvest crop residue is a fine fuel that burns directly in the field and mostly by flaming in many
mechanized agricultural systems. In contrast, when crops are harvested by hand the residue is often
burned in large piles that may smolder for weeks.
This seasonal crop residue burning season typically occurs in the spring and summer seasons
and also occasionally occurs in the autumn and winter. Pollution gases, dust, and suspended particle
matters resulting from crop residue burning emissions result in poor air quality that threaten human
health and terrestrial ecosystems. The Chinese presidential decree included the prohibition of crop
residue burning into the Law of the People's Republic of China on the Prevention and Control of
Atmospheric Pollution in August 2015 (http://www.chinalaw.gov.cn, last access on 17 July 2019 ),
and since then the crop residue burning events were banned throughout China. Therefore, we obtain
a decrease in fire numbers in China since 2015.
In addition, the El Niño Southern Oscillation (ENSO) can cause large scale variations in the
convection, circulation, and air temperature of the global atmosphere-ocean system (Liu et al., 2017;
Zhao et al., 2002), which could affect the distribution, frequency, and intensity of biomass burning
emissions (Schaefer et al., 2018). Furthermore, ENSO could also alter the destruction processes of
tropospheric species through their photochemical reactions with tropospheric OH (Zhao et al., 2002).
Zeng et al. (2002) found that the abnormally enhancement of tropospheric CO and HCN observed
in northern Japan in 1998 were associated with the 1997–1998 ENSO events (Zhao et al., 2002).
The large correlation between ENSO and HCN at Lauder (45°S) revealed a detectable ENSO
influence on biomass burning (up to 51 % – 55 %) (Schaefer et al., 2018; Zeng et al., 2012).
Presumably, the significant enhancements between September 2015 and July 2016 for tropospheric
CO and HCN columns at Hefei and most selected NDACC stations were also related to the 2015 –
2016 ENSO events.
**6 Conclusion**
The first multiyear measurements of HCN in the polluted troposphere in densely populated
eastern China have been presented here. Tropospheric HCN column amounts were derived from
solar spectra recorded with ground-based high spectral resolution Fourier transform infrared (FTIR)
spectrometer at Hefei (117°10′E, 31°54′N) between 2015 and 2018. The seasonality and interannual
variability of tropospheric HCN columns in eastern China have been investigated. The potential
sources that drive the observed HCN seasonality and interannual variability were determined by
using the GEOS-Chem tagged CO simulation, the global fire maps and the PSCFs (Potential Source
Contribution Function) calculated using HYSPLIT back trajectories.
The tropospheric HCN columns over eastern China showed significant seasonal variations with
three monthly mean peaks throughout the year. The magnitude of the tropospheric HCN peak in
May > September > December. The tropospheric HCN column reached a maximum of $(9.8 \pm 0.78)$
$\times 10^{15}$ molecules/cm$^2$ in May and a minimum of $(7.16 \pm 0.75) \times 10^{15}$ molecules/cm$^2$ in November.
In most cases, the tropospheric HCN columns at Hefei (32°N) are higher than the NDACC FTIR
observations. Enhancements of the tropospheric HCN columns were observed between September
2015 and July 2016 compared to the counterpart measurements in other years. The magnitude of
the enhancement ranges from 5 to 46% with an average of 22%. Enhancement of tropospheric HCN
($\Delta$HCN) is correlated with the coincident enhancement of tropospheric CO ($\Delta$CO), indicating that
enhancements of tropospheric CO and HCN were due to the same sources.
The GEOS-Chem tagged CO simulation, the global fire maps and the PSCFs analysis revealed
that the seasonal maxima in May is largely due to the influence of biomass burning in South Eastern
Asia (SEAS) ($41 \pm 13.1\%$), Europe and Boreal Asia (EUBA) ($21 \pm 9.3\%$) and Africa (AF) ($22 \pm$
$4.7\%$). The seasonal maxima in September is largely due to the influence of biomass burnings in





EUBA (38 ± 11.3%), AF (26 ± 6.7%), SEAS (14 ± 3.3%) and NA (13.8 ± 8.4%). For the seasonal
maxima in December, dominant contributions are from AF (36 ± 7.1%), EUBA (21 ± 5.2%), and
NA (18.7 ± 5.2%).
The enhancements of both tropospheric HCN and CO columns between September 2015 and
July 2016 at Hefei (32°N) were attributed to an elevated influence of biomass burnings in SEAS,
EUBA, and Oceania (OCE) in this period. Particularly, an elevated fire numbers in OCE in the
second half of 2015 dominated the tropospheric HCN enhancement in September – December 2015.
An elevated fire numbers in SEAS in the first half of 2016 dominated the tropospheric HCN
enhancement in January – July 2016.
***Data availability.*** The CO and HCN measurements at the selected NDACC sites can be found by
the link http://www.ndaccdemo.org, and the CO and HCN measurements at Hefei are available on
request.
***Author contributions.*** YS conceived the concept and prepared the paper with inputs from all
coauthors. CL, WW, CS, HY, XX, MZ, and JL carried out the experiments. The rest authors
contributed to this work via provide refined data or constructive comments.
***Competing interests.*** The authors declare that they have no conflict of interest.
***Acknowledgements.*** This work is jointly supported by the National High Technology Research and
Development Program of China (No.2019YFC0214800, No.2017YFC0210002, No.
2016YFC0203302), the National Science Foundation of China (No. 41605018, No. 41405134,
No.41775025, No. 41575021, No. 51778596, No. 91544212, No. 41722501, No. 51778596), and
Outstanding Youth Science Foundation (No. 41722501). The processing and post processing
environment for SFIT4 are provided by National Center for Atmospheric Research (NCAR),
Boulder, Colorado, USA. The NDACC network is acknowledged for supplying the SFIT software
and the HCN and CO data. The LINEFIT code is provided by Frank Hase, Karlsruhe Institute of
Technology (KIT), Institute for Meteorology and Climate Research (IMK-ASF), Germany. The
MeteoInfo software is provided by Prof. Yaqiang Wang, Chinese Academy of Meteorological
Sciences. The authors acknowledge the NOAA Air Resources Laboratory (ARL) for making the
HYSPLIT transport and dispersion model available on the Internet. The Mauna Loa (20°N) FTIR
site is operated by National Center for Atmospheric Research (NCAR), U.S.A., and the Lauder
(45°S) and Arrival Heights (78°S) sites are operated by National Institute of Water & Atmospheric
Research, New Zealand. The multi-decadal monitoring program of ULiege at the Jungfraujoch
station has been primarily supported by the F.R.S.-FNRS and BELSPO (both in Brussels, Belgium)
and by the GAW-CH programme of MeteoSwiss. The International Foundation High Altitude
Research Stations Jungfraujoch and Gornergrat (HFSJG, Bern) supported the facilities needed to
perform the FTIR observations.
***Appendix.***

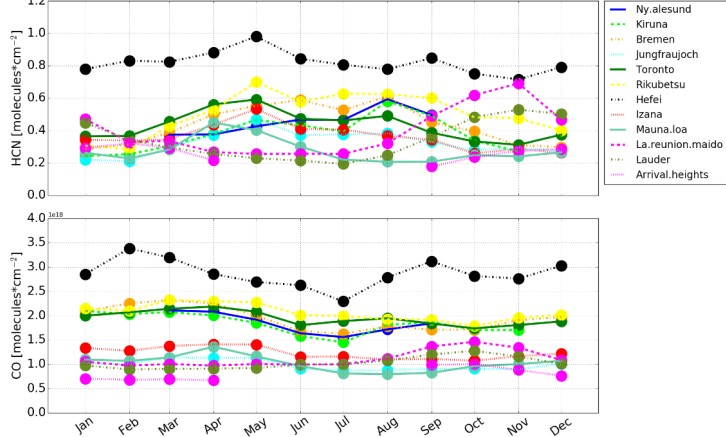

Fig. A1. Monthly means of the tropospheric CO and HCN columns at Ny Alesund, Kiruna, Bremen, Jungfraufoch,
Toronto, Rikubetsu, Hefei, Izana, Mauna Loa, La Reunion Maido, Lauder, and Arrival Heights from 2015 to 2018.



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
