# Peer review of "FTIR time series of tropospheric HCN in eastern China: seasonality, interannual variability and source attribution"

_Atmospheric Chemistry and Physics, 2019_

## Referee Comment (RC1) · Anonymous Referee #1 · 14 Feb 2020

The study by Sun et al reports the column and partial columns of HCN, a known biomass indicator, along with the column CO from Heifei in China. The authors compare their data with other similar measurements from the NDACC community, and use the standard agreed protocol for retrieve these columns. They derive emission ratios of HCN with respect to CO, and also use back trajectory analyses to trace where the airmasses originated. The trajectories from various global sectors are discussed and compared in the context of the seasonality in the HCN to CO enhancements, which the authors conclude is consistent with the same sources. Finally global fire counts are used to show that fires from various sources explain the observed seasonality in the HCN column enhancements.

[Figure]

In general the paper is well written, organised, and follows NDACC recognised analysis procedures. The skill level of this group Heifei is very good, and improves all the time. The paper is therefore recommended for publication in ACP subject to the list of mainly minor corrections, but also a few not so minor points for the authors to correct/address.

Corrections:

1) P2 line 41: replace "in polluted troposphere in eastern China at middle latitude ..." with "in the polluted troposphere over eastern China at middle latitudes ..."

2) P2/3 line 57/1: suggest replacing last sentence on page 2 with "Both HCN and CO are regularly measured at Hefei ($32°$N) using the FTIR observations, where influences from biomass burning occurring at long distances or locally can be assessed."

3) P3, line 12: "... evaluation of global .." => "...evaluation of the global ..."

4) P3, line 30/31: spelt out the "&"

5) P3 line33: The field stop is employed to maximise the S/N consistent with the maximum frequency possible for the wavenumber range that is selected. Is this is what adapt means? Explain.

6) P3 caption of figure 1: what is significant about these star spots? Refer to the text for an explanation (just a simple, see section xx). Even so it is not clear on page 16 where this is mentioned, exactly why these points were chosen. Are they the middle of the selected source regions?

7) P5, fig 2: Why show the averaging kernel plot of HCN with these two layers when the actual partial column used in this study is from 0 to 15 km? HCN has a dof of 1.3 so there are not 2 independent layers, unlike CO. Perhaps it would be better to plot two averaging kernels for HCN, the total column and the 0-15 km layer?

8) P5 table 2: I am not sure about the tabulated error for the line intensity for CO and HCN, they seem to be the wrong way around? The reported uncertainty for the line

intensity in hitran2008 for CO and HCN is 2-5% and 5-10% respectively. So why is the reported % error for the CO line intensity higher than HCN?

9) P6 line 16: Combine these sentences to read "…September > December, while for the tropospheric CO column, ..."

10) P6 line 18: perhaps "phase" is a better word than "timing"? This word is used elsewhere as well.

11) P7, line 6: type in the word "Fig. A"

12) P7, line 27: be more specific about what timeframes you are actually comparing, what exactly do you mean by "counterpart"?

13) P7, line 32: combine these sentences to read "…2016, while…"

14) P7/8: Maybe summarise all this information in this paragraph in a table? It is otherwise tedious for any reader.

15) P9, line 2,: "proceed" is not the correct word here. Should be something like "using a linear ..."

16) P10, line 28: "seasonal" => "seasonally"

17) P10, line 31: "…occurred before the timing of tropospheric HCN enhancements within one month period. Then, we" replace with "occurring before the timing of tropospheric HCN enhancements within a one month period. Finally, we …"

18) P13, fig 8 caption: What is the ratio in fig 8a, what is being ratioed against what?

19) P13, fig 8 caption, line 5: Do you mean the each red dot is a fire over the 10 day period. What is written here implies a fire that occurred 10 days before. So this is cumulative over the 10 days, yes?

20) P14, fig 9b: These plots are really hard to make out. Perhaps explore another way of plotting so it clearly shows the different datasets. My suggestion is to try different

plotting symbols as the one you use seem too large.

21) P15, line 13, "Particularly," => "In particular, "

22) P16, line 8: "cause"

23) P16, line 15: "lifetimes ..." => "where lifetimes ..."

24) P17, line 7: These numbers from other studies though should be entered as a range in table 5, that is, a column that says "literature values" to give the reader a sense of where the emission ratios in this study fall.
* * *

---

## Referee Comment (RC2) · Anonymous Referee #2 · 23 Feb 2020

The authors presented a multi-year timeseries (2015-2018) of HCN partial columns observed at a densely populated Chinese city, Hefei, using a ground-based high spectral resolution Fourier transform infrared (FTIR) spectrometer. CO columns were also presented for comparison. The authors identified the seasonal maxima of tropospheric HCN columns at this site and compared their seasonality and interannual variations to the HCN columns obtained from other NDACC sites. Drivers of HCN seasonality and interannual variations at Hefei were determined through GEOS-Chem tagged CO simulations, as well as the global fire maps and a back-trajectory analysis technique. They also used CO measurements to estimate the HCN/CO emission enhancement ratios to quantify biomass burning emissions.

[Figure]

This version (manuscript-version2) has been significantly improved from the original submission in terms of analysis and presentation. The results are valuable to the community. I suggest some minor corrections, which are detailed below. The authors should perform a thorough proof-reading of the manuscript to further improve the presentation in order to meet the publishing standards of ACP.

1) Page 1, Line 35: Are these values multi-year mean or are they the maximum values throughout the whole period? Please clarify.

2) p2, line 34-35: Could you also point out where these regions are located in China?

3) p2, line 53: add "where" before "influences from biomass . . ."

4) p3, line 6: replace "can improve" with "aims to improve"

5) p3, line 17: replace "it is planned to" with "plan to"

6) p3, line 20: replace "a highest" with "the highest"

7) p3, line 22: "to ensure a higher signal to noise . . . and a faster. . ." compared to what?

8) p3, line 26-27: can you spell out "&" as appropriate here.

9) P4, line 8: change to "Whole-Atmosphere Community Climate Model (WACCM) v6"

10) P4, line13-15: spell out all species names when they first appear.

11) P4, line 24: did you calculate the tropopause height from the NCEP reanalysis data? If so, replace "deduced" with "calculated".

12) P5, line 19: please spell out GEOS-FP.

13) P6, line 5: the order should be "Global Fire Assimilation System (GFAS) v1.2"

14) P6, line 9: change to "period of"

15) P6, line 11: Please be consistent in placing acronyms. Should be "the Multi- resolution Emission Inventory for China (MEIC)".

16) P6, line 18-19: It is confusing here, as the GEOS-Chem OH shown in Shindell et al. (2006) is lower than the multi-model mean (Table 1). Could you clarify?

17) P6, line 31-33: this sentence can be modified as "In this study, we only investigate the influence from the biomass burning sources. The regional definition..."

18) P6: at the beginning of section 2.3, could you briefly say the purpose of using PSCF in this study? Just add some context.

19) P7, line 37: delete "total" before "troposphere"

20) P8, line 6-10: these few lines can be re-organised to make the statements clearer. For example, "Note that the largest seasonal peak of HCN occurs in May which is 3 months later than that of CO which occurs in February, but the other two seasonal peaks for both species occur in the same months, i.e., in September and December respectively. Otherwise, their seasonal cycles show similarities."

21) P10, line 11-17: could you elaborate a bit more on possible mechanisms why correlations between CO and HCN are negative or weak at these sites, especially at Bremen? Have you checked the seasonal correlation between CO and HCN at Bremen?

22) P10, line 24: how about "...is an effective quantity to identify biomass burning emissions..."?

23) P10, line 31: should be "indicating that these..."

24) P10, line 32: change to "throughout the year"

25) P11, line 1: change to "is located ..., therefore emissions..."

26) P14, line 22: change to "both the first and the second half of the year"

27) P14, line 24: change to "in the second half of the year"

28) P14, line 25: same as above "in the first half of the year"

29) P14, line 27: change to "As shown in Figs..."

30) P14, line 34: replace "Drives" to "Driven"

31) P15, line 3: "meteorological conditions"

32) P15, line 7-8: change to "through large-scale atmospheric circulation"

33) P17, line 2: replace "driven" with "drove"

34) P17, first paragraph: Are these elevated fire counts related to ENSO? Can you discuss the possible ENSO link here, rather than at the end of the section?

35) P17, lines 28: should it be "Zhao et al. (2000) or (2002)" here?

36) P17, line 30-31: please re-phrase this sentence as "There is a close correlation between ENSO and HCN columns at Lauder (45oS) (Zeng et al., 2012; Schaefer et al., 2018), and Schaefer et al. (2018) quantified a detectable ENSO influence on biomass burning of up to 51-55%".

37) P17, line 32: replace "presumably" with "very likely".

38) P17, line 37: delete "amounts" and "s" to "HCN column"

39) In Conclusion, could you provide some further remarks on the long-term outlook of FTIR observations at this site.

---

## Author Comment (AC1) · 11 Mar 2020

Response to Referee #1:

Thanks very much for your comments, suggestions and recommendation with respect to improve this paper. The response to all your comments are listed below.

General comments :

The study by Sun et al reports the column and partial columns of HCN, a known biomass indicator, along with the column CO from Heifei in China. The authors compare their data with other similar measurements from the NDACC community, and use the standard agreed protocol for retrieve these columns. They derive emission ratios of HCN with respect to CO, and also use back trajectory analyses to trace where the air masses originated. The trajectories from various global sectors are discussed and compared in the context of the seasonality in the HCN to CO enhancements, which the authors conclude is consistent with the same sources. Finally global fire counts are used to show that fires from various sources explain the observed seasonality in the HCN column enhancements.

In general the paper is well written, organised, and follows NDACC recognised analysis procedures. The skill level of this group Heifei is very good, and improves all the time. The paper is therefore recommended for publication in ACP subject to the list of mainly minor corrections, but also a few not so minor points for the authors to correct/address.

**Response:** This paper has been subjected to a revision based on the comments from two referees. All your comments are appreciated and have been addressed in the revised version. Main changes/improvements are listed as follows:

Specific comments:

1) P2 line 41: replace "in polluted troposphere in eastern China at middle latitude . . ." with "in the polluted troposphere over eastern China at middle latitudes . . ."

2) P2/3 line 57/1:   suggest replacing last sentence on page 2 with "Both HCN and CO are regularly measured at Hefei (32N) using the FTIR observations, where influences from biomass burning occurring at long distances or locally can be assessed."

3) P3, line 12: ". . . evaluation of global .." => ". . .evaluation of the global . . ."

4) P3, line 30/31: spelt out the "**&**"

**Response:** We have done above corrections in the revised version

5) P3 line33: The field stop is employed to maximise the S/N consistent with the maximum frequency possible for the wavenumber range that is selected. Is this is what adapt means? Explain.

**Response:** We have changed this sentence to "The entrance field stop size ranging from 0.80 to 1.5 mm was employed to maximise the SNR consistent with the maximum frequency possible for the selected wavenumber range."

6) P3 caption of figure 1: what is significant about these star spots? Refer to the text for an explanation (just a simple, see section xx). Even so it is not clear on page 16 where this is mentioned, exactly why these points were chosen. Are they the middle of the selected source regions?

**Response:** We have included a new Figure 1 in the revised paper, where the star spots are replaced by six rectangles for tagged CO definitions. It looks more clear than the old ones. Please check.

7) P5, fig 2: Why show the averaging kernel plot of HCN with these two layers when the actual partial column used in this study is from 0 to 15 km? HCN has a dof of 1.3 so there are not 2 independent layers, unlike CO. Perhaps it would be better to plot two averaging kernels for HCN, the total column and the 0-15 km layer?

**Response:** We have done this as your suggestion and plotted two averaging kernels for HCN, the total column and the 0-15 km layer.

8) P5 table 2: I am not sure about the tabulated error for the line intensity for CO and HCN, they seem to be the wrong way around? The reported uncertainty for the line intensity in hitran2008 for CO and HCN is 2-5% and 5-10% respectively. So why is the reported % error for the CO line intensity higher than HCN?

**Response:** In the error analysis, we assume *a priori* error covariance of line intensity as 5% for both CO and HCN. The error in table 2 randomly selected from all measurements shown that the resulting % error for the CO line intensity in the troposphere is higher than HCN.

Some remarks should be clarify: 1, the error analysis may site, measurement, and

layer dependent. For the selected retrieval, the tropospheric AK uncertainties of CO used for error analysis (Rodgers, 2000) are larger than HCN. 2, the error analysis were performed after retrieval, the error in line intensity estimation for either CO or HCN would not impact the time series and thus would not alter the point of this paper. In the revised version, we still set *a priori* error covariance of line intensity to 5% for both CO and HCN, and don't make any revision.

9) P6 line 16: Combine these sentences to read ". . .September > December, while for the tropospheric CO column, ..."

10) P6 line 18: perhaps "phase" is a better word than "timing"? This word is used elsewhere as well.

**Response:** We have done above corrections in the revised version

11) P7, line 6: type in the word "Fig. A"

**Response:** We have moved this figure to supplement. Now Fig.A1 is Fig. S2 in the supplement.

12) P7, line 27: be more specific about what timeframes you are actually comparing, what exactly do you mean by "counterpart"?

**Response:** In the revised version, the "counterpart" has been replaced by "concurrent". The timeframes of NDACC selected for comparison are the same as that of Hefei. This sentence has been changed to " Enhancements of both tropospheric HCN and CO columns between September 2015 and July 2016 at Hefei (32°N) were observed compared to the measurements in other years."

13) P7, line 32: combine these sentences to read ". . .2016, while. . ."

**Response:** We have done this in the revised version.

14) P7/8: Maybe summarise all this information in this paragraph in a table? It is otherwise tedious for any reader.

**Response:** We have condensed all this information to "The enhancements of both tropospheric HCN and CO columns within the same period were also observed at the selected NDACC stations except Ny Alesund (79°N) and Kiruna (68°N). The winter enhancements were not shown over Ny Alesund (79°N) and Kiruna (68°N) because of the polar night in the Arctic which interrupted the observations in winter. The

magnitude of the enhancement in tropospheric HCN column at the selected NDACC stations between September 2015 and July 2016 ranges from 3 to 213%, and for CO ranges from 4 to 62%."

15) P9, line 2,: "proceed" is not the correct word here. Should be something like "using a linear ..."

16) P10, line 28: "seasonal" => "seasonally"

17) P10, line 31: ". . .occurred before the timing of tropospheric HCN enhancements within one month period. Then, we" replace with "occurring before the timing of tropospheric HCN enhancements within a one month period. Finally, we . . ."

**Response:** We have done above corrections in the revised version.

18) P13, fig 8 caption: What is the ratio in fig 8a, what is being ratioed against what?

**Response:** In the revised version, this figure has been replaced by Fig. 11.

19) P13, fig 8 caption, line 5: Do you mean the each red dot is a fire over the 10 day period. What is written here implies a fire that occurred 10 days before. So this is cumulative over the 10 days, yes?

**Response:** Correct. In the revised version, this figure has been replaced by Fig. 11.

20) P14, fig 9b: These plots are really hard to make out. Perhaps explore another way of plotting so it clearly shows the different datasets. My suggestion is to try different plotting symbols as the one you use seem too large.

**Response:** In the revised version, this figure has been replaced by Figs. 13 and 14. Now it should be more clear.

21) P15, line 13, "Particularly," => "In particular, "

**Response:** We have done this in the revised version.

22) P16, line 8: "cause"

23) P16, line 15: "lifetimes ..." => "where lifetimes ..."

24) P17, line 7: These numbers from other studies though should be entered as a range in table 5, that is, a column that says "literature values" to give the reader a sense of where the emission ratios in this study fall.

**Response:** We have removed this paragraph in the revised version. Instead, a tagged CO simulation was included as response to referee # 2's request. Please check.

---

## Author Comment (AC2) · 11 Mar 2020

Response to Referee #2:

Thanks very much for your comments, suggestions and recommendation with respect to improve this paper. The response to all your comments are listed below.

General comments :

The authors presented a multi-year timeseries (2015-2018) of HCN partial columns observed at a densely populated Chinese city, Hefei, using a ground-based high spectral resolution Fourier transform infrared (FTIR) spectrometer. CO columns were also presented for comparison. The authors identified the seasonal maxima of tropospheric HCN columns at this site and compared their seasonality and interannual variations to the HCN columns obtained from other NDACC sites. Drivers of HCN seasonality and interannual variations at Hefei were determined through GEOS-Chem tagged CO simulations, as well as the global fire maps and a back-trajectory analysis technique. They also used CO measurements to estimate the HCN/CO emission enhancement ratios to quantify biomass burning emissions.

This version (manuscript-version2) has been significantly improved from the original submission in terms of analysis and presentation. The results are valuable to the community. I suggest some minor corrections, which are detailed below. The authors should perform a thorough proof-reading of the manuscript to further improve the presentation in order to meet the publishing standards of ACP.

**Response:** This paper has been subjected to a revision based on the comments from two referees. We also perform a thorough proof-reading of the manuscript to further improve the presentation. All your comments are appreciated and have been addressed in the revised version. Main changes/improvements are listed as follows:

Specific comments:

1) Page 1, Line 35: Are these values multi-year mean or are they the maximum values throughout the whole period? Please clarify.

**Response:** They are monthly means based on multi-year measurements. We have clarified this in the revised version.

2) p2, line 34-35: Could you also point out where these regions are located in China?

**Response:** In the revised version, we have included a map of China showing the three

most developed regions—Jing-Jin-Ji (Beijing-Tianjin-Hebei), Yangtze River Delta (including Shanghai), and Pearl River Delta (including Guangzhou and Hong Kong–Shenzhen) in Fig. S1. It looks more clear than put these regions in the world map in Figure 1.

3) p2, line 53: add "where" before "influences from biomass . . ."

4) p3, line 6: replace "can improve" with "aims to improve"

5) p3, line 17: replace "it is planned to" with "plan to"

6) p3, line 20: replace "a highest" with "the highest"

**Response:** The above corrections have been done in the revised version.

7) p3, line 22: "to ensure a higher signal to noise . . . and a faster. . ." compared to what?

**Response:** To avoid misleading, we have changed it to " However, all mid-infrared (MIR) spectra are recorded with a spectral resolution of 0.005 cm$^{-1}$ to follow NDACC convention".

8) p3, line 26-27: can you spell out "&" as appropriate here.

**Response:** We have done this in the revised version.

9) P4, line 8: change to "Whole-Atmosphere Community Climate Model (WACCM) v6"

**Response:** We have done this in the revised version.

10) P4, line13-15: spell out all species names when they first appear.

**Response:** We have done this in the revised version.

11) P4, line 24: did you calculate the tropopause height from the NCEP reanalysis data? If so, replace "deduced" with "calculated".

**Response:** Yes, and we have done this in the revised version.

12) P5, line 19: please spell out GEOS-FP.

**Response:** We have done this in the revised version.

13) P6, line 5: the order should be "Global Fire Assimilation System (GFAS) v1.2"

**Response:** We have done this in the revised version.

14) P6, line 9: change to "period of"

**Response:** We have done this in the revised version.

15) P6, line 11: Please be consistent in placing acronyms. Should be "the Multi-resolution Emission Inventory for China (MEIC)".

**Response:** We have done this in the revised version.

16) P6, line 18-19: It is confusing here, as the GEOS-Chem OH shown in Shindell et al. (2006) is lower than the multi-model mean (Table 1). Could you clarify?

**Response:** We have consulted the GEOS-Chem team at Harvard university during the ACPD phase. The developer told me that what we actually used in this paper was "monthly mean OH concentrations archived from a previous full-chemistry simulation" rather than that used from Shindell et al. (2006). In the revised version, we have changed this sentence to " The OH fields were obtained from monthly mean OH concentrations archived from a previous full-chemistry simulation."

17) P6, line 31-33: this sentence can be modified as "In this study, we only investigate the influence from the biomass burning sources. The regional definition. . ."

**Response:** We have done this in the revised version.

18) P6: at the beginning of section 2.3, could you briefly say the purpose of using PSCF in this study? Just add some context.

**Response:** We have done this in the revised version.

19) P7, line 37: delete "total" before "troposphere"

**Response:** We have done this in the revised version.

20) P8, line 6-10: these few lines can be re-organised to make the statements clearer. For example, "Note that the largest seasonal peak of HCN occurs in May which is 3 months later than that of CO which occurs in February, but the other two seasonal peaks for both species occur in the same months, i.e., in September and December respectively. Otherwise, their seasonal cycles show similarities."

**Response:** We have done this in the revised version.

21) P10, line 11-17: could you elaborate a bit more on possible mechanisms why correlations between CO and HCN are negative or weak at these sites, especially at Bremen? Have you checked the seasonal correlation between CO and HCN at Bremen?

**Response:** Actually, here are the overall correlations by using all measurements from

2015 to 2018. We have doubled check the programing script for this figure, and found that all fittings are correct. We have included more explanations in the revised paper, i.e., "Biomass burning is the dominant source of HCN and industrial emissions only contribute additional minor sources (Bange and Williams, 2000; Holzinger et al., 1999; Lobert et al., 1990). In contrast, anthropogenic, biomass burning, $CH_4$ and NMVOC oxidations are major sources of CO, and their contributions are season and location dependent. Therefore, the correlation between HCN and CO tropospheric column is also season and site dependent. High correlation of these two species is supposed to be observed if biomass burning dominates the CO variability, and vice visa."

22) P10, line 24: how about ". . .is an effective quantity to identify biomass burning emissions. . ."?

23) P10, line 31: should be "indicating that these. . ."

24) P10, line 32: change to "throughout the year"

25) P11, line 1: change to "is located . . ., therefore emissions. . ."

26) P14, line 22: change to "both the first and the second half of the year"

27) P14, line 24: change to "in the second half of the year"

28) P14, line 25: same as above "in the first half of the year"

29) P14, line 27: change to "As shown in Figs. . ."

30) P14, line 34: replace "Drives" to "Driven"

31) P15, line 3: "meteorological conditions"

32) P15, line 7-8: change to "through large-scale atmospheric circulation"

33) P17, line 2: replace "driven" with "drove"

**Response:** We have done above corrections in the revised version.

34) P17, first paragraph: Are these elevated fire counts related to ENSO? Can you discuss the possible ENSO link here, rather than at the end of the section?

**Response:** We have moved the last paragraph here and analyzed the possible ENSO link. "Many studies have revealed that the El Niño Southern Oscillation (ENSO) can cause large scale variations in the convection, circulation, and air temperature of the global atmosphere-ocean system (Liu et al., 2017; Zhao et al., 2002), which could affect the distribution, frequency, and intensity of biomass burning emissions

(Schaefer et al., 2018). Furthermore, ENSO could also alter the destruction processes of tropospheric species through their photochemical reactions with tropospheric OH (Zhao et al., 2002). Zhao et al. (2002) found that the abnormally enhancement of tropospheric CO and HCN observed in northern Japan in 1998 were associated with the 1997–1998 ENSO events (Zhao et al., 2002). There is a close correlation between ENSO and HCN columns at Lauder (45°S) (Zeng et al., 2012; Schaefer et al., 2018), and Schaefer et al. (2018) quantified a detectable ENSO influence on biomass burning of up to 51-55% (Schaefer et al., 2018; Zeng et al., 2012). Very likely, the elevated fire counts which caused significant enhancements between September 2015 and July 2016 for tropospheric CO and HCN columns at Hefei and most selected NDACC stations were related to the 2015 – 2016 ENSO events."

35) P17, lines 28: should it be "Zhao et al. (2000) or (2002)" here?

**Response:** We have changed it to Zhao et al (2002).

36) P17, line 30-31: please re-phrase this sentence as "There is a close correlation between ENSO and HCN columns at Lauder (45ºS) (Zeng et al., 2012; Schaefer et al., 2018), and Schaefer et al. (2018) quantified a detectable ENSO influence on biomass burning of up to 51-55%".

37) P17, line 32: replace "presumably" with "very likely".

38) P17, line 37: delete "amounts" and "s" to "HCN column"

**Response:** We have done above corrections in the revised version.

39) In Conclusion, could you provide some further remarks on the long-term outlook of FTIR observations at this site.

**Response:** In the revised version, we have included the sentence "Most high resolution FTIR instruments are located in Europe and Northern America, whereas the number of sites in Asia, Africa, and South America is very sparse. As one of few FTIR stations on Asian continent, the long-term observations of trace gases at Hefei are crucial to understand global warming, regional pollution, long term transport, and contribute to the evaluation of satellite data and model simulations."

---

## Author Response (AR2)

**NOTE: This file includes two sections. Section 1 presents comments from editor, the corresponding point-by-point responses, and the related changes in the manuscript. Section 2 is the marked-up manuscript.**

**Section 1:** (The black font are comments from editor, and the red font are authors' responses as well as the related change clarifications)

**(1) Detailed response to comments from editor:**

I have reviewed the revised version of your manuscript. Over all, I find that you have considered the comments of the reviewers properly. I have some (minor) comments myself, that I list in the following. The language of the paper will need some copy-editing that will be done in any case. I have not checked the language carefully for this reason. Please consider my comments and resubmit a revised version of your manuscript (with marked changes), together with a list of changes and replies to my comments.

**Response:** Thanks very much for your comments. All your comments have been considered and addressed seriously in the revised version. Point by point responses and changes/improvements are listed as follows. We have tried our best to improve the language of the paper, and we hope the copy-editing service by Copernicus can further improve the language.

Comments to the revised version (Page and line numbers refer to the version of the manuscript included in the replies to the reviewers):

Section 2.1.3. " … HCN averaging kernels only have one maxima at 10 km. Both CO and HCN retrievals ." => "… HCN averaging kernels have only one maximum at 10 km. Both CO and HCN retrievals …"

**Response:** We have done this in the revised version.

Remove "We can see in Table 2, …" => " The typical degrees of freedom (DOFS) obtained at Hefei (32°N) over the total atmosphere for CO and HCN are about 2.8 ± 0.3 (1σ) and 1.3 ± 0.2 (1σ), respectively (Table 2)."

**Response:** We have done this in the revised version.

Figs. 3,4, Table 4: The name of the station is Jungfraujoch, and it is separated as Jungfrau-joch

**Response:** We have done this in the revised version.

page 20: line 3: … and their relative contributions …

**Response:** We have done this in the revised version.

end of line 5: vice versa

**Response:** We have done this in the revised version.

page 22, starting in line 3:

" … follows. First, the GEOS-Chem CO VMR profiles of all tracers in the grid box containing the Hefei (32°N)site were converted to partial column profiles and linearly interpolated and regridded onto the FTIR vertical retrieval grid. This was necessary in order to account for the differences in the vertical levels of the model and the FTIR (Barret et al., 2003).Then, The GEOS-Chem CO partial column profiles are smoothed by the normalized FTIR CO total column averaging kernel following Rodgers and Connor (2003)."

In my opinion, the reversed order of steps would make more sense: First smooth the vertically well resolved model profiles with the retrieval averaging kernels; this provides you with the profile the instrument/retrieval scheme would have seen it if the model profile was the truth. Then calculate the partial columns in the same way as calculated from the observations.

I strongly suggest you do a test to quantify the difference between the two approaches.

**Response:** Actually, what we really done is "First smooth the vertically well resolved model profiles with the retrieval averaging kernels and then calculate the partial columns in the same way as calculated from the observations". As described in Rodgers and Connor (2003), this is a pretty standard method to compare results from different systems. We have modified previous misleading descriptions to " First, the GEOS-Chem CO VMR profiles of all tracers in the grid box containing the Hefei (32°N) site were linearly interpolated and regridded onto the FTIR vertical retrieval grid. This was necessary in order to account for the differences in the vertical levels of the model and the FTIR (Barret et al., 2003). Then, the interpolated GEOS-Chem CO profiles were smoothed by the FTIR CO averaging kernel following Rodgers and Connor (2003). Finally, we compared the partial columns calculated from the

smoothed GEOS-Chem CO profiles with the FTIR ones." in the revised version.

Fig. 8/9/10: Indicate in the Fig. caption that these data belong to Hefei alone.

**Response:** We have indicated "at Hefei (32°N)" in captions of Figs. 8/9/10.

Fig. 11: what are the red marks in the maps?

**Response:** The are fire locations/numbers accumulated from the FIRMS fire atlas. Each red mark represent a fire there. We have included the statement "FIRMS fire numbers are shown in red dots for 13, 7, 10, and 14 days prior to the arrival time, respectively." in the revised version.

Fig. 12: What is the unit of the color scale?

**Response:** They are conditional probability values without unit (see section 2.3). The higher the value (or the closer the value to 1.0), the higher the sensitivity.

Page 24, second-but-last para: I do not quite understand the following sentence:

"Driven by the Asian monsoon anticyclone (AMA), the dominant circulation feature in the Indian–Asian upper troposphere–lower stratosphere region during the Asian monsoon, these emissions can be transported to eastern China which contributed to the tropospheric HCN peak in May. "

Do you indicate here that the May peak observed in Hefei is linked to the AMA? I don't believe so; the Asian monsoon just starts in May, and the most pronounced transport patterns occur in July/August.

**Response:** In the revised version, we have changed "Driven by the Asian monsoon anticyclone (AMA), the dominant circulation feature in the Indian–Asian upper troposphere–lower stratosphere region during the Asian monsoon, these emissions can be transported to eastern China which contributed to the tropospheric HCN peak in May." to "Largely driven by deep convection followed by northward transport into the mid-latitude westerlies (Liu et al., 2003), these emissions can be transported to eastern China which contributed to the tropospheric HCN peak in May." .

The following reference that support this change has been included.

Liu, H., Jacob, D., J., Bey, I., Yantosca, R., M., Duncan, B., N.: Transport pathways for asian pollution outflow over the pacific: interannual and seasonal variations. Journal of Geophysical Research, 108(D20), 8786, 2003.

page 26, line -8: abnormally => abnormal

**Response:** We have done this in the revised version.

**Section 2:** marked up file, as follows

In briefly, we have revised the paper based on editor's comments, and tried our best to improve the language problem.The marked up file is as follow, please check the red sentences for details:

[revised manuscript text omitted]